# Saharan dust long-range transport across the Atlantic studied by an airborne Doppler wind lidar and the MACC model

Fernando Chouza[1], Oliver Reitebuch[1], Angela Benedetti[2], Bernadett Weinzierl[3]

[1]Institut für Physik der Atmosphäre, Deutsches Zentrum für Luft- und Raumfahrt (DLR), Oberpfaffenhofen, Germany
[2]European Centre for Medium-Range Weather Forecasts, Reading, UK
[3]Faculty of Physics, University of Vienna, Boltzmanngasse 5, A-1090 Wien, Austria

*Correspondence to*: F. Chouza (fernando.chouza@dlr.de)

**Abstract.** A huge amount of dust is transported every year from North Africa into the Caribbean region. This paper presents an investigation of this long-range transport process based on airborne Doppler wind lidar (DWL) measurements conducted during the SALTRACE campaign (June-July 2013), as well as an evaluation of the ability of the MACC global aerosol model to reproduce it and its associated features. Although both the modelled winds from MACC and the measurements from the DWL show a general good agreement, some differences, particularly in the African Easterly Jet (AEJ) intensity, were noted. The observed differences between modelled and measured wind jet speeds are between 5 and 10 m s$^{-1}$. The vertical aerosol distribution within the Saharan dust plume and the marine boundary layer is investigated during the June-July 2013 period based on the MACC aerosol model results and the CALIOP satellite lidar measurements. While the modelled Saharan dust plume extent shows a good agreement with the measurements, a systematic underestimation of the marine boundary layer extinction is observed.

Additionally, three selected case studies covering different aspects of the Saharan dust long-range transport along the West African coast, over the North Atlantic Ocean and the Caribbean are presented. For the first time, DWL measurements are used to investigate the Saharan dust long-range transport. Simultaneous wind and backscatter measurements from the DWL are used, in combination with the MACC model, to analyse different features associated with the long-range transport, including an African Easterly Wave trough, the African Easterly Jet and the Intertropical Convergence Zone.

## 1 Introduction

Every year huge amounts of Saharan dust originating from North Africa is transported across the Atlantic into the Caribbean region and the north of South America. The transport, mainly occurring during the summer season, starts with the uplifting of dust by turbulent convection and low level winds with high speed (Bou Karam et al., 2008), being the amount of emitted dust regulated by different factors like the soil humidity and vegetation. Once lofted, the dust is dispersed into a deep mixed layer, reaching altitudes of up to 6 km during summer (Messager et al., 2010). The dominating easterly winds west advect the dust laden air masses, which are undercut by the cooler and moister air from the marine boundary layer (MBL) as they

reach the West African coast, forming an elevated layer of relative warm, dry and dust laden air called Saharan Air Layer (SAL). As the SAL leaves the African continent, its lower and upper bounds are defined by a strong inversion at approximately 1.5 km and a relative weaker inversion at around 5 km to 6 km, respectively (Prospero and Carlson, 1972; Karyampudi et al., 1999).

Along its life cycle, the airborne dust interacts with the environment in different ways. During its long range transport phase, the dust modifies the radiative budget, acts as cloud and ice nuclei and is observed to modify the cloud glaciation process (e.g. Seifert et al., 2010). Various mechanisms have been proposed to explain the controversial influence of the SAL on the evolution of African Easterly Waves (AEW) into tropical storms (e.g. Dunion et al., 2004; Evan et al., 2006; Lau and Kim, 2007). Regularly mineral dust is impacting aviation, in particular in regions near dust sources, by inducing poor visibility

(Weinzierl et al., 2012). As it deposits, the Saharan dust can affect the air quality (Prospero, 1999) and serves as source of nutrients for plankton and the Amazon basin (Yu et al., 2015).

Several studies were conducted during the last years to provide further insight in the previously mentioned processes, including field campaigns and long term studies based on models, satellite, airborne, and ground based measurements. Among others, we can mention the African Monsoon Multidisciplinary Analysis (AMMA) and NASA-AMMA (NAMMA)

campaigns (Zipser et al., 2009) conducted in 2006, focused on the analysis of the AEW and its evolution into tropical cyclones, and the Saharan Mineral Dust Experiments 1 and 2 (SAMUM-1 and SAMUM-2) (Heintzenberg, 2009; Ansmann et al., 2011) conducted in 2006 and 2008, respectively, which were designed to investigate the Saharan dust size distribution and morphology, and the relation with the optical and radiative properties in the West African region. Additionally, during April-May 2013, a shipborne lidar onboard the research vessel Meteor conducted a transatlantic cruise between the

Caribbean and the west coast of Africa in order to characterize the mixtures of the Saharan dust with biomass burning aerosols and evaluate the change in its optical properties as result of the long range transport process (Kanitz et al., 2014). While field campaigns provide an intensive set of observations in relative small regions and time intervals, satellite observations provide regular observations with a limited set of parameters. On the other hand, global and regional models are frequently used in dust long-range transport studies and forecasting (e.g. Schepanski et al., 2009; Kim et al., 2014; Gläser et

al., 2015), as they provide valuable information to interpret the observational data collected by campaigns and satellite measurements. The performance evaluation of these models, like during the intercomparison initiative AeroCom (Aerosol Comparison between Observations and Models) (Kinne et al., 2003), is then not only vital to improve the models, but also to know the limits and accuracy of the conclusions extracted based on these models.

The Saharan Aerosol Long-range Transport and Aerosol-Cloud-Interaction Experiment (SALTRACE; Weinzierl et al.,

2016) performed in June/July 2013 was framed in this context. SALTRACE was planned as a closure experiment to investigate the Saharan dust long-range transport between Africa and the Caribbean, with focus on the dust aging and deposition processes and the characterization of its optical properties. The campaign dataset include a set of measurements from the ground-based aerosol lidars BERTHA (Backscatter Extinction lidar-Ratio Temperature Humidity profiling Apparatus) (Haarig et al., 2015) and POLIS (Portable Lidar System) (Groß et al., 2015), in-situ and sun photometer

instruments deployed on Barbados (main SALTRACE super-site), Cape Verde and Puerto Rico and airborne aerosol and wind measurements from the DLR (Deutsches Zentrum für Luft- und Raumfahrt) research aircraft Falcon similar to the measurements done during SAMUM (Weinzierl et al., 2009 and 2011). For the first time, an airborne Doppler wind lidar (DWL) was deployed to study the dust transport across the Atlantic Ocean, including its interaction with island induced

gravity waves (Chouza et al., 2016).

In this study, this unique set of DWL wind and extinction measurements along the main dust transport acquired during SALTRACE are used, in combination with dropsondes and CALIPSO (Cloud–Aerosol Lidar and Infrared Pathfinder Satellite Observations) satellite measurements, to analyse the Saharan dust long-range transport and to evaluate the performance of the MACC (Monitoring Atmospheric Composition and Climate) aerosol global model to reproduce different

associated atmospheric features, like the African Easterly Jet (AEJ) and its interaction with the SAL, the AEWs and the ITCZ (Intertropical Convergence Zone), among others. Such a comparison provides not only an insight about the current model capabilities, which is of great relevance for model based studies, but the opportunity to identify the model weaknesses and provide a starting point for future improvements. This type of evaluation has never been performed with a datasets that includes both meteorological fields as well as atmospheric composition fields. Although there is no direct feedback from the

atmospheric composition fields onto the meteorological fields (temperature and winds) as the run was not coupled through interactive radiative processes, the aerosols are transported and advected by the winds. Hence an evaluation of both winds and aerosols in the SAL is a very effective way to understand how well the model represents this natural phenomenon.

The paper is organized as follows. Section 2 presents a brief description of the datasets used for this study, including an evaluation of the DWL accuracy based on collocated dropsonde measurements. Section 3 provides a comparison between

the DWL and CALIPSO measurements with the backscatter and winds from the MACC model in the West African and Caribbean regions for the time period of the SALTRACE campaign. Section 4 presents three case studies with relevant features of the Saharan dust long-range transport. Finally, summary and relevant conclusions are presented in Section 5.

## 2 Observations and model data

During the SALTRACE campaign, the DLR Falcon research aircraft conducted 31 research flights between 10 June and 15

July 2013, with most of the flights concentrated close to the West African coast and the Caribbean (Fig. 1). The DLR Falcon was equipped with a set of instruments for in-situ particle measurements, a DWL and dropsondes. Ground based measurements were conducted in Cape Verde, Puerto Rico and Barbados. Two aerosol lidars and sun photometers were installed on the west coast of Barbados, at the Caribbean Institute for Meteorology and Hydrology (CIMH) (13°08'55" N 59°37'30" W, 110 m ASL), while a sun-photometer and a set of ground based in-situ instruments were deployed at Ragged

Point, Barbados (Kristensen et al., 2016). A complete list of the instruments involved in the campaign can be found on the SALTRACE website (http://www.pa.op.dlr.de/saltrace/instruments.html) and an overview of SALTRACE is given in

Weinzierl et al., (2016). The following subsections describe in detail the characteristics of the datasets used for this study, including the used fields from the MACC model.

## 2.1 Dropsondes

A set of 34 Vaisala RD93 dropsondes, operated in conjunction with the NCAR AVAP System (Busen, 2012), were launched from the Falcon during SALTRACE, providing vertical profiles of temperature, relative humidity and wind speed with a vertical resolution of approximately 10 m. According to the manufacturer, the horizontal wind measurements have an accuracy of 0.5 m s$^{-1}$ RMS (Root-Mean-Square) (Vaisala, 2009), where this accuracy definition incorporates both the systematic and the random error of the measurements. Figure 1 indicates the geographical position at which each dropsonde was launched.

## 2.2 Doppler wind lidar

The coherent DWL deployed on the DLR Falcon 20 research aircraft during SALTRACE is based on a CLR Photonics instrument (Henderson et al., 1993) combined with a two wedge scanner and acquisition system developed at DLR (Köpp et al., 2004; Reitebuch, 2012). The system operates at a wavelength of 2.02254 μm, with a pulse duration full width at half maximum (FWHM) of 400 ns, a pulse energy of 1-2 mJ, and a repetition frequency of 500 Hz.

When mounted on an aircraft, the system can be operated in two modes: the conical scanning and the nadir pointing mode. The conical scanning mode consists of 24 lines of sight (LOS) in a conical distribution with an off-nadir angle of 20° and a staring duration of 1 s per LOS direction. The inversion algorithm (Smalikho, 2003; Weissmann et al., 2005) is then applied to the 24 LOS measurements from each scan to retrieve horizontal wind speed vector with a horizontal resolution of around 6-10 km (depending on the aircraft speed) and a vertical resolution of 100 m. On the other hand, the nadir pointing mode can be used to retrieve vertical wind measurements with a horizontal resolution of approximately 200 m (Chouza et al., 2016). Both modes allow, by means of an adequate calibration, the retrieval of backscatter and extinction coefficients with a horizontal resolution of approximately 200 m and a vertical resolution of 100 m (Chouza et al., 2015). As in the case of CALIOP, the backscatter and extinction retrievals form the DWL measurements require to assume a lidar ratio for each aerosol type. For the retrievals presented in this study, a lidar ratio of 55 sr was used for the Saharan dust, while 30 sr and 35 sr were used for the marine boundary layer and the marine-dust mixed layer respectively. To determine the accuracy of the DWL backscatter and extinction retrieval, the DWL profiles were compared to measurements from the ground-based aerosol lidar POLIS and the CALIPSO satellite, exhibiting a good agreement, with systematic differences lower than 20% in the backscatter coefficient retrievals.

During SALTRACE, the DWL totalized 75 hours of measurements, from which 56 hours were performed in scanning mode. The atmospheric conditions present during most of the campaign flights, characterized by a large dust load between ground and 4 to 6 km, provided excellent backscatter conditions for the DWL.

In order to evaluate the accuracy of the horizontal wind speed and direction measurements performed by the DWL, a set of vertical profiles were compared to those retrieved from collocated dropsondes observations. Only DWL profiles with less than one minute of difference with respect to the dropsonde launch time were included in the comparison, while the dropsonde measurements were vertically averaged to match the DWL vertical resolution. The results, summarized in Figs. 2 and 3, indicate a systematic difference (bias) of 0.08 ms$^{-1}$ and a standard deviation of 0.92 ms$^{-1}$ for the wind speed difference between the DWL and the dropsondes measurements. The bias and the standard deviation does not exhibit significant dependence with the measurement altitude or measured wind speed, while the amount of points available for comparison as a function of altitude illustrates the average DWL coverage observed during SALTRACE. The mean wind speed measured by the dropsondes and the DWL exhibit a maximum between 4 km and 6 km, associated with the presence of the AEJ (African Easterly Jet). For the case of the wind direction, the mean difference is 0.5° and the standard deviation is 10°, with values of around 5° between ground and 6 km, and higher values between 6 km and 8 km. The mean direction values between ground and 6 km are between 90° and 100°, compatible with the easterly dust transport direction. The DWL performance evaluation shows results which are generally consistent to those obtained in the North Atlantic region during the Atlantic THORPEX Regional Campaign (A-TReC) in November 2003 (Weissmann et al., 2005), with slightly higher standard deviation for the wind direction measurements and smaller differences in the wind speed for the SALTRACE dataset.

## 2.3 CALIOP

The Cloud–Aerosol Lidar with Orthogonal Polarization (CALIOP), the primary instrument of the CALIPSO satellite, is a two-wavelength polarization-sensitive lidar launched in 2006 by NASA (Winker et al., 2009). Based on a three channel receiver, one for backscatter measurements at 1064 nm and two for the parallel and cross-polarized backscatter at 532 nm, the lidar is able to provide aerosol type classification, aerosol optical depth (AOD) and extinction coefficient vertical profiles. This allows a characterization of the mean Saharan dust vertical distribution close to the source and in the Caribbean region during the SALTRACE campaign, as well as an evaluation of the model to reproduce it. The dust vertical distribution is a key parameter which has a direct influence in the radiative transfer calculations and the atmospheric stability.

For this study, the Level 2 (V3.3) dataset was used (https://www-calipso.larc.nasa.gov/search/). This dataset includes aerosol type classification, total column cloud, aerosol and stratospheric optical depth, and extinction profiles with a vertical resolution of 60 m and a horizontal resolution of 5 km. In order to include only the most accurate measurements retrieved by CALIOP, a series of masks were applied to the data. First, profiles for which the clouds and stratospheric optical depth where not zero were masked out. A second mask was applied to keep only bins with the "Volume description Bit" equal to 0 ("clean air") or 3 ("aerosols"), cloud-aerosol discrimination (CAD) score <-80 and quality control (QC) flag equal to 0 or 1. Finally, the AOD was derived from the vertical integration of extinction profiles where no data points were missing.

Since the CALIOP extinction coefficient retrieval relies on assumed lidar ratios, any systematic difference in the lidar ratio will directly affect the derived extinction profiles. An estimation of the systematic errors associated with the CALIOP extinction retrieval in the Saharan dust transport region is presented in Tesche et al. (2013). Based on a comparison between a ground-based Raman lidar deployed on the Cabo Verde region during SAMUM-2 and different CALIOP overpasses, this

study concluded that CALIOP extinction profiles retrievals of the SAL exhibit an average systematic underestimation of around 15% during summer season, while in some cases the difference reached 30 %. This systematic underestimation of CALIOP extinction retrievals can be explained by the fact that the CALIOP retrieval scheme assumes a dust lidar ratio of 40 sr, while Raman lidar measurements conducted during SAMUM-2 (Groß et al., 2011) and SALTRACE (Groß et al., 2015) indicate values close to 55 sr.

According to Wandinger et al. (2010), the dust lidar ratio of 40 sr used by the CALIOP algorithms is an effective value which takes in account the effect of multiple scattering and leads to backscatter coefficient profiles which are in good agreement with ground-based lidar measurements. Nevertheless, the use of this effective dust lidar for the calculation of the SAL extinction coefficients leads to their systematic underestimation. Although different techniques to correct this systematic error were proposed (e.g. Wandinger et al., 2010; Amiridis et al., 2013), the CALIOP retrievals shown in this

study are presented in their original form. In cases where this effect could alter the derived conclusions, relevant comments are included (Sec. 3.3).

## 2.4 The MACC model

As part of the formerly Global Monitoring for Environment and Security (GMES) initiative (now Copernicus), intended to improve our understanding of the environment and climate change, the European Centre for Medium-Range Weather

Forecasts (ECMWF) developed the MACC model, a forecasting and reanalysis system for aerosols, greenhouse gases and reactive gases based on the assimilation of satellite and in-situ observations (Hollingsworth et al., 2008). This project extends the capabilities of the operational ECMWF Integrated Forecast System (IFS) by including a new set of forecasted variables. A detailed description of the model and parameterizations is given in Morcrette et al. (2009), while the assimilation process is explained in Benedetti et al. (2009) and a companion paper by Mangold et al. (2011) provides a first validation of the

model results. Additional validation of the aerosol MACC products in northern Africa and Middle East can be found in the work by Cuevas et al. (2015).

The MACC aerosol parametrization is based on the LOA/LMD-Z (Laboratoire d'Optique Atmosphérique/Laboratoire de Météorologie Dynamique-Zoom) model (Boucher et al., 2002; Reddy et al., 2005) and includes five types of tropospheric aerosols: dust, sea salt, organic, black carbon and sulfate. The first two corresponds to natural sources and are represented in

the model by three size bins for dust (0.03-0.55 µm, 0.55-0.9 µm and 0.9-20 µm) and other three for sea salt (0.03-0.5 µm, 0.5-5 µm and 5-20 µm). Currently, stratospheric aerosols are based on the climatology already included in the operational IFS as no prognostic fields are included in the model.

The sea salt emission parametrization is based on the model 10 m winds and a source function based on the works by Guelle et al. (2001) and Schulz et al. (2004). The emission, calculated for a relative humidity of 80%, is integrated for the three sea salt model bins previously described. Emissions of dust depend on the 10-m wind, soil moisture, the UV-visible component of the surface albedo and the fraction of land covered by vegetation when the surface is snow-free, adapted from Ginoux et al. (2001). A correction of the 10-m wind to account for gustiness is also included (Morcrette et al. 2008). The model includes aerosol transport by diffusion, convection and advection treated with a semi-lagrangian approach.

A set of different removal processes are included in the model, namely, dry deposition, sedimentation and wet deposition. Wet and dry deposition were directly adapted from the LOA/LMD-Z model, while the sedimentation scheme follows the work from Tompkins (2005a) on ice sedimentation.

One of the distinctive characteristics of the MACC model is the full integration of the aerosol model to the forecasting model. The 4-D Var system regularly employed in the IFS was extended to assimilate AOD observations from the moderate-resolution imaging spectroradiometer (MODIS) on Terra and Aqua satellites at 550 nm, including the total aerosol mixing ratio as additional control variable (Benedetti et al., 2009). This assimilation process is based on the adjustment of the model control variables (e.g., initial conditions) in order to minimize a cost function. This function depends on the difference between the observations and its model equivalent, and the relative weight assigned to each of them based on the estimated observational and model uncertainties. In the case of the MODIS AOD assimilation, the observations are compared with a model derived AOD. The AOD observation operator derives the optical depth based on precomputed aerosol optical properties and model relative humidity for the aerosol species included in the model. After assimilation, the model output represents the best statistical compromise between the model background (forecast running without assimilation) and observations.

The MACC reanalysis was not available for 2013, having stopped in 2012, hence the operational run at the time of the campaign (June-August 2013) was used in this study. The model run with a time resolution of 3 h at T255L60, which corresponds to a horizontal resolution of approximately 0.8° x 0.8° (78 km x 78 km) and 60 vertical levels, with the top at 0.1 hPa. The assimilation window is 12 h. Among others, the MACC model provides as output vertically resolved total AOD (natural + anthropogenic aerosols) corresponding to each of the layers bounded by the model levels. Since this study focuses on the comparison of the model with lidar observations, extinction coefficient profiles were calculated dividing the layer AOD by the layer thickness.

## 3 MACC model evaluation

The wind profile is a key parameter which determines the of the Saharan dust long-range transport. Although many studies were conducted to address the specific role of the trade winds, the AEJ (African Easterly Jet), and the African Easterly Waves (AEWs) in the dust lifting, transport and deposition, many questions remains still open (Ansmann et al., 2011; Schulz et al., 2012). As the Saharan dust transport path is mainly over the North Atlantic Ocean, the number of assimilated wind

observations into a global model is very limited, which in turn affects their accuracy. Since many long-range transport studies use wind information provided by models, the evaluation of the accuracy of these simulations is of great importance. In this context, the wind data set acquired by the DWL during SALTRACE provides a unique opportunity to evaluate the wind fields by the MACC model which are ultimately responsible for the Saharan dust long range transport.

The assimilation of AOD form MODIS into the MACC model constraints the total aerosol mass, which is a great advantage considering the large uncertainties associated with the dust emission quantification (Huneeus et al., 2011). Nevertheless, since the assimilated variable is a column integrated measurement, no independent constraints are introduced on each aerosol type and their vertical distributions. For that reason, the ability of the model to simulate the vertical distribution of each aerosol type still has to be evaluated. While CALIPSO provides regular backscatter and extinction coefficient measurements which can be used to evaluate the model output and eventually provide data for assimilation, the aerosol detection sensitivity

of CALIOP is limited compared to ground based or airborne lidars. The DWL extinction profiles, simultaneously retrieved with the horizontal wind measurements, allow an independent evaluation of the simulated aerosol vertical distribution as well as the possibility to study the interaction between wind and aerosol distributions.

Since most of the flights were conducted either close to the West coast of Africa or in the Caribbean region, two regions

were defined to study the model results with the DWL and CALIOP dataset. The first region is located at the west coast of Africa [0-30N;10-30W], while the second region encloses the flights in the Caribbean [0-30N;50-70W].

### 3.1 Method

Because the MACC model provides an output every 3 hours, the horizontal resolution is approximately 80 km (T255) and the vertical levels are not homogeneously distributed as a function of altitude, a re-binning and interpolation process is

necessary to compare the model output with the DWL and CALIOP observations. First, the observational data is re-binned to a grid with a horizontal resolution of 80 km and a vertical grid defined according to the 60 model levels for the U.S. Standard Atmosphere, 1976 (http://www.ecmwf.int/en/forecasts/documentation-and-support/60-model-levels). Only bins with more than 50% of the expected amount of values are used, otherwise, the bin is filled with a missing value flag. Then, the MACC model is linearly interpolated in space and time to match the re-binned observation data grid. Once that

observations and model data are in a comparable grid, different comparison techniques were applied. In the case of the horizontal winds, the re-gridded DWL measurements and model output corresponding to flights in each region were studied to evaluate the ability of the model to reproduce the main features of the wind in these two regions.

In the case of the CALIOP extinction retrievals, two months of measurements corresponding to June and July 2013 were included in the comparison due to the relative low amount of AOD and extinction retrievals available after the quality check

process (Sec. 2.3). The measurements of CALIOP inside the regions defined in Sec. 3 and the corresponding MACC profiles were then, after re-gridding, zonally averaged in each region to provide a statistically relevant result. Extinction coefficient and AOD are reported by CALIOP and MACC at 532 nm.

### 3.2 Horizontal wind

The spatially averaged DWL wind retrievals were compared with the interpolated MACC winds in the West Africa and Caribbean regions defined in Sec. 3. In the case of the West African region, a total of 7 flights conducted between 11 and 17 June 2013 are included in the comparison, which corresponds to 1202 speed/direction pairs. In the Caribbean region, 13 measurement flights conducted between 20 June 2013 and 11 July 2013 are compared with a total of 1532 speed/direction measurement pairs. In order to investigate possible correlations between the altitude and speed and the differences between the lidar and the model, the mean and standard deviation of the difference between the DWL and MACC as a function of the altitude, the number of compared measurements and the mean speed and direction corresponding to the African and Caribbean regions are shown in Fig. 4.

In the case of the West African region, Figs. 4c and 4g indicate a good agreement in the profile shape between the mean measured and simulated wind speed and direction respectively, with the altitude of the wind maxima and minima being well reproduced by the model. Regarding the wind speed magnitude, Fig. 4d indicates an underestimation of the wind speed by the model between 0.5 km and 6 km, with a maximum difference of 5 ms$^{-1}$ at around 5 km. This is approximately coincident with the altitude of the maximum simulated (13.2 ms$^{-1}$) and measured wind speeds (17 ms$^{-1}$), which corresponds to the presence of the AEJ in the region. The underestimation of the AEJ by the ECMWF model was already reported in previous studies, like the one conducted during the JET2000 campaign in August 2000 (Thorncroft et al., 2003; Tompkins et al., 2005b). The comparison, performed against a set of dropsonde measurements, indicated a similar difference profile, with a maximum difference of around 5 ms$^{-1}$. As in the case of the comparison between the lidar and the dropsondes, the distribution of points included in the comparison (Fig. 4j) is representative of the lidar coverage, which in turn is determined by the aerosol load distribution and cloud coverage. Most of the measurements are available below 6 km, in coincidence with the SAL upper bound.

The left row of Fig. 4 presents the results for the Caribbean region. In contrast to the previous case, some differences in the mean wind speed and direction structures are recognizable in Figs. 4a and 4e. Both, the DWL and the model show a similar behavior below 0.8 km, characterized by an increasing wind speed as function of altitude, reaching its maximum at 0.8 km. Although the simulated and measured wind gradients are similar, the simulated speeds are around 1 ms$^{-1}$ larger than the measured ones. Above the boundary layer, the model shows a decrease in the wind speed and a general underestimation of the wind speed compared to the DWL measurements. As in the previous case, the distribution of compared points as function of the altitude (Fig. 4i) is correlated with the dust load and cloud coverage. The strong reduction of measurement points above 4 km is related with the general decrease in the altitude of the SAL top boundary as the dust moves westward.

### 3.3 Aerosol extinction coefficient

In order to analyze the accuracy of the MACC model vertical aerosol distribution, the extinction coefficient profiles were compared to those retrieved by CALIOP during the months of June and July 2013 for a wavelength of 532 nm. An overview

of the comparison is presented in Fig. 5, where the zonal mean of the simulated and measured AOD and extinction coefficient is presented, together with the difference between the model and measurement, for the West African and Caribbean regions indicated in Fig. 1.

Figures 5a and 5b show the zonal mean of the AOD for the two regions. The simulated and measured AOD close to the African coast (Fig. 5b) exhibit a similar bell shape, with the model maxima located at 18.5° N and the weighted mean of the CALIOP AOD at 15°N. In the Caribbean region, the model and the CALIOP measurements show a better agreement, with the center of the dust plume located at 16°N in both cases. The change in the AOD between both regions, a reduction by half due to the transport between the African coast and Barbados, is in relative good agreement. Nevertheless, the AOD north and south of the dust plume show consistent higher values in the model. This bias is consistent with the study presented in Kim et al. (2013), which showed that MODIS AOD values corresponding to clean marine aerosols approximately doubles the AOD reported by CALIOP.

While a relative good agreement of the measured and simulated AOD is expected due to the constraints introduced by the assimilation of MODIS measurements, the vertical distribution of the dust, which plays a crucial role in atmospheric radiative transfer studies (Zhang et al., 2013), has to be evaluated. The zonal averaged extinction coefficient measured by CALIOP in West Africa (Fig. 5d) shows an elevated dust plume between 5°N and 30°N, with its lower bound at 1 km and its upper bound at approximately 6 km. Below 1 km, an aerosol loaded boundary layer is recognizable, with higher extinction coefficients below the SAL, probably due to dust entrainment into the atmospheric boundary layer (ABL). In the Caribbean, the dust plume top sinks to around 4 km, and the extinction decreases to about half of the values measured close to the source region. As in the previous case, the ABL shows higher extinction below the SAL than north or south of it; the characteristic wedge shape as we move from higher to lower latitudes can also be noted.

The modelled aerosols vertical distribution close to the dust source region is presented, together with the difference with the CALIOP measurements, in Figs. 5f and 5h respectively. The modelled dust plume shows relative good agreement in spatial distribution with respect to the CALIOP measurements, extending between 10°-30°N. Coincident with the AOD measurements presented in Fig. 5b, the simulated plume is slightly displaced in north direction and exhibits a higher maximum extinction at around 2 km. This displacement, clearly visible in the difference plot, could lead to a change in the interaction between the SAL and the AEJ, which in turn will promote different dust transport patterns. However, the major difference between CALIOP and MACC is visible in the boundary layer and above the SAL. In the ABL, the model strongly underestimates the extinction coefficient. The model average extinction in the ABL (< 1 km) is 0.045 $km^{-1}$ in the Caribbean region and 0.065 $km^{-1}$ in the West Africa region, while the CALIOP average retrievals are 0.145 $km^{-1}$ and 0.13 $km^{-1}$, respectively. Although the CALIOP retrievals of the ABL extinction are affected by the uncertainty in the lidar ratios and the underestimation of the SAL extinction, these effects cannot explain the large discrepancy observed between the model and the CALIOP retrieval.

The underestimation of the SAL extinction due to an underestimation of the dust lidar ratio would lead to an underestimation of the CALIOP backscatter in the boundary layer, which in turn would lead to an even larger CALIOP retrieval. On the other

hand, and since the aerosol load found in the ABL below the SAL is dominated by a mixture of marine aerosol and dust, the expected average CALIOP lidar ratio will oscillate between the lidar ratio values corresponding to pure marine (20 sr) and pure dust (40 sr). In the case of the retrievals presented in Fig. 5c and 5d, the average lidar ratio used by CALIOP for the calculation of the ABL (< 1 km) extinction coefficient was found to be between 33 sr in the West African coast region and

37 sr in the Caribbean region. These values are similar to those presented in Groß et al. (2016), where the average lidar ratio observed in the ABL was 26±5 sr, with values ranging from 22±5 sr in the case of pure marine aerosol to 35±3 sr for the case of a dust-dominated ABL. These results, corresponding to June-July 2013, are in agreement with the CALIOP retrievals presented in Cuevas et al., (2015), where a similar difference in the ABL extinction in the West African region (M'Bour) was also observed during the summer seasons of 2007 and 2008.

The comparison of the CALIOP extinction retrievals and the corresponding model results indicate a relative small overestimation of the SAL extinction by the model. Even if the CALIOP extinction retrievals of the SAL are multiplied by a correction factor of 1.375 (55 sr/40 sr) to take in account the underestimation of the lidar ratio by CALIOP discussed in Sec. 2.3, most of the MACC results are still above the CALIOP retrievals (not shown). The SAL AOD derived from MACC is still larger than the SAL AOD derived from CALIOP extinction. Based on a SAL lower and upper bounds between 2 and

5 km in the Caribbean and between 1 and 6 km in the African region, the SAL AOD values based on MACC are 50% and 10% larger than the CALIOP retrievals, respectively. Additionally, and although the pure dust lidar ratio used for CALIOP is 40 sr, sometimes the algorithm identifies part of the SAL as polluted dust, with an associated lidar ratio of 55 sr. This leads to an average lidar ratio larger than 40 sr, which in turn leads to a higher extinction and to a smaller systematic bias. If instead of correcting the CALIOP extinction retrievals by using the correction factor suggested by Wandinger et al. (2010)

the mean underestimation reported by Tesche et al. (2013) from Table 5 is used (1/0.858=1.165), the differences are even larger.

Above the SAL, the model indicate the presence of very thin aerosol layer extending up to 20 km which is not visible in the CALIOP measurements. Since the extinction coefficients predicted by the model above the SAL are in the order of the sensitivity threshold of CALIOP (~0.01 km$^{-1}$), a direct validation of these features is not possible only by means of CALIOP

measurements. Recent studies (e.g. Rogers et al., 2014) indicate that the lack of detection of weakly backscattering aerosol layers in the free troposphere leads to an underestimation of approximately 0.02 in the CALIOP column AOD. This effect can partially explain the systematically lower CALIOP AOD values observed in Figs. 5a and 5b. In order to further investigate this features, a comparison of the model with the DWL measurements performed during the transfer flight between Cape Verde and South America is presented in the next section.

**4 Characteristics of the Saharan dust long-range transport**

The study presented in the previous section provided a general overview of the performance of the model. In this section, a set of three selected flights form the SALTRACE campaign are used to provide a deeper insight into some characteristic

features of the Saharan dust long range transport mechanism. The first case study, corresponding to a flight between Cape Verde and Dakar, is included to further investigate the differences observed in the AEJ intensity as well as the ability of the model to reproduce the generation of AEWs. The second case study presents measurements over the North Atlantic Ocean, which provides a second comparison of the AEJ speed and the opportunity to analyze the interaction of the dust with the ITCZ. Finally, the third case presents the measurements of a dust outbreak in the Caribbean region after the long-range transport across the North Atlantic.

In order to facilitate the interpretation of these case studies, the corresponding synoptic situation is presented in Fig. 6. These charts, which include the wind direction and wind speed at 700 hPa and the total AOD, are based on the MACC model results. Additionally, in order to situate the case studies within the context of the SALTRACE campaign, a Hovmöller diagram with the MACC meridional winds and AOD averaged between 0°N and 30°N is presented in Fig. 7. This diagram simplifies the visualization of some features associated with the Saharan dust long-range transport, like the dust outbreaks, the AEWs and the passage of the tropical storm Chantal. A period of 3 to 5 days between outbreaks and a transport time of around 5 days between Dakar and Barbados can be easily noted in the AOD diagram (Fig. 7, right). In order to highlight the strong correlation of these outbreaks with the propagation of AEWs, the wave troughs relevant for the presented case studies are indicated in Fig. 7.

## 4.1 Dust plume at the West African coast

On 12 June 2013 the Falcon performed a research flight at the West coast of Africa, departing from Cape Verde at 08:52 UTC and landing in Dakar at 12:08 UTC. The model winds at 700 hPa are presented, together with the AOD, in Fig. 6a. Wind speeds around 15 ms$^{-1}$ to 25 ms$^{-1}$ can be observed between 10°N and 15°N, which is compatible with the presence of the AEJ. The inverted V-shape disturbance in the AEJ flow in the Dakar region, suggests the passage of an AEW during the observation period. The observation of the change in the meridional wind flow direction at pressure levels between 850 hPa and 700 hPa is a typical way to detect AEWs, with the wave trough defined as the point of zero meridional wind (change from northerly to southerly flow) (e.g. Reed et al., 1977). According to this definition, the wave trough can be recognized at 17°W during the observation period (AEW 1, Fig. 7). This wave, sampled during the flight on 12 June 2013, propagated further to the west at an approximately speed of 8° (~900 km) per day, reaching Barbados on 17 June 2013. Several other waves can be also observed in the presented period. All wave cases propagate westward with a similar speed, which gives place to a mean travel time of around 5 days between the African west coast and Barbados. This result is consistent with observations and previous studies on the AEWs behavior (Zipser et al., 2009).

The AOD for the Dakar region presented in Fig. 6a shows a dust loaded air mass on the leading edge of the AEW and a region of less dust load behind the trough. Satellite images (not shown), indicate the presence of strong convective activity behind the trough, which is a typical feature of the AEWs (Fink et al., 2003; Cifelli et al., 2010). The propagation of the dust outbreak as it leaves the West African coast can be observed in Fig. 7 (right). According to the model results, this outbreak was the largest one (highest AOD) during the SALTRACE observation period. The modelled passage of the AEW seen in

Fig. 6a is coincident with a reduction in the amount of exported dust behind the trough (AEW 1). This could be the result of the enhanced wet deposition associated with the convective cell activity (Desboeufs et al., 2010).

The DWL measurements corresponding to the selected case study are presented in Fig. 8. The extinction coefficient profiles derived from the DWL measurements presented in Fig. 8a are in qualitative agreement with the distribution simulated by the model (Fig. 8b), with an elevated dust plume between 1 km and 6 km riding on top of the marine boundary layer as it leaves the African continent (17°W). These results are compatible with the BERTHA ground-based measurements carried out in Cabo Verde during the SAMUM-2b campaign (Tesche et al., 2011) in summer 2008, where a 0.5–1.0 km deep maritime boundary layer topped by a 4–5 km deep mineral dust layer was typically observed. As has been noted in the previous section, the model underestimates the extinction of the marine boundary layer and overestimates the extinction of the SAL.

The average SAL extinction retrieved from the DWL for altitudes between 1 km and 6 km and longitudes between 16.17° W and 15° W is 0.11 km$^{-1}$, while the corresponding MACC average extinction is 0.4 km$^{-1}$. This difference is much larger than the systematic error of ±20% estimated for the DWL backscatter retrievals. Although in this case the compared values are extinction coefficients and not backscatter coefficients, the lidar ratios used in the DWL retrieval (55 sr for the SAL, 35 sr for the marine-dust mixed layer, and 30 sr for the marine boundary layer) are in agreement with those found in the literature and small local variations cannot explain the large difference observed in this case study.

The enhanced cloud coverage associated with the previously mentioned convective system can be seen starting at 9:35 UTC (behind the wave trough) as white regions in the DWL extinction plot (Fig. 8a). These clouds, visible at altitudes between 6 and 8 km, completely blocked the DWL laser, which in turn led to missing data below them (white regions). An associated decrease in the aerosol load in this clouds region can also be noted in the DWL retrievals starting at 9:47 UTC. Since the MACC model extinction product does not include clouds, the enhanced cloud coverage cannot be seen in Fig. 8b.

The horizontal wind profiles retrieved by the DWL and simulated by MACC are presented as speed-direction and u-v components in Figs. 8c-j. The presence of the AEJ can be recognized on both: the DWL speed measurements (Fig. 8c) and the corresponding MACC speed profiles (Fig. 8d) for altitudes between 3 km and 6 km. The magnitude of the jet is, nevertheless, strongly underestimated by the model by almost 10 ms$^{-1}$, which is above the mean difference of 5 ms$^{-1}$ observed in the previous section for the mean of all flights in the AEJ region. The underestimation of the AEJ impacts not only in the amount of transported Saharan dust, but also in the propagation and development of AEWs (Leroux and Hall, 2009). Below 2 km, a land-sea breeze system located over Dakar (-15° N; 18° W) can be recognized in the DWL and MACC wind speed-direction profiles. The passages of the AEWs are easier to recognize when the wind vector is presented as u and v components. The u component of the measured (Fig. 8g) and simulated (Fig. 8h) wind profiles is dominated by the AEJ, while the v component captures the wave trough observed in Figs. 7 and 8. Both, the position of the trough and the amplitude of the wind meridional component are well reproduced by the model. A strong coincidence between the dust plume border and trough is also visible in both cases.

## 4.2 Dust long-range transport across the Atlantic

The second case corresponds to a flight between Cape Verde and South America performed on 17 June 2017 between 13:24 UTC and 17:21 UTC. This flight gives the opportunity to study the interface between the SAL and the ITCZ, as well as the AEJ, the Tropical Easterly Jet (TEJ) (Chen and Loon, 1987) and the ability of the model to reproduce them. The synoptic situation, presented in Figs. 6b and 9, shows a dust plume laterally bounded by an AEW trough located at 15° W (AEW 2, Fig. 7) and an AEW crest located at 40° W, and south bounded by the ITCZ, located between 3°N and 8°N. In the DWL measurements, the ITCZ can be recognized by the presence of strong convective activity and the associated development of convective clouds at 6 km (white areas) and between 2°N and 7°N.

The DWL and MACC extinction and horizontal wind profiles are presented in Fig. 9. The MACC model is able to reproduce the main characteristics of the aerosol vertical distribution. North of the ITCZ, a dust plume with its upper bound at 6 km is visible by both, the model and the DWL. In the case of the DWL measurements, the SAL is lower bounded by a characteristic wedge-shaped marine boundary layer topped by low level clouds, while in the case of the MACC model, the lower bound seems to be lower and constant as function of the latitude. The model and dropsonde temperature and water vapor mixing ratio vertical profiles presented in Fig. 10 show a general good agreement, although a consistent underestimation of the model temperature of around 2 °K below 1 km can be observed. In the case of the first dropsonde (DS 1), the temperature and mixing ratio profiles are compatible with the differences observed by the DWL, with the model showing an ABL top inversion approximately 500 m below the measured one. Although in the case of the DWL the coverage in the ITCZ is limited by the presence of clouds, the dust load south of it is clearly reduced by the effect of wet deposition on both, the DWL and the MACC model simulation. As in the previous section, the model indicates the presence of aerosols above the SAL, especially in the ITCZ region, which is not captured by the lidar. Since the extinction coefficients shown by the model are quite low, an additional set of extinction plots in logarithmic scale were added to highlight this feature (Figs. 7c and 7d). The average extinction coefficient shown in Fig. 7d for an altitude of 9.2 km range from 0.003 $km^{-1}$ between 14:35 and 14:50 UTC to 0.01 $km^{-1}$ between 15:05 to 15:35 UTC. Because the DWL relies on the atmospheric aerosols for the retrieval of wind measurements, such change in the aerosol load should result in a change in the wind retrieval coverage. Nevertheless, this is not observed in Figs. 7e and 7g, where the wind retrieval coverage is limited to a band of approximately 2 km below the Falcon, independently of the geographical location. While the extinction coefficients shown by MACC above the SAL are normally below 0.01 $km^{-1}$, its detection by the DWL, especially close to the Falcon, would be expected in the case of a real feature. The aerosols shown by the model in the upper troposphere are likely to be an artifact introduced by the model in order to compensate the lack of extinction in the ABL and hence balance the assimilated AOD from MODIS.

The horizontal wind speed profiles are presented in Figs. 7e and 7f for the DWL and the MACC model, respectively. The AEJ, visible in the DWL measurements on the north of the ITCZ for altitudes 2 km and almost 7 km can also be seen in the model, although slightly displaced to the north and with a lower mean speed. Below 1.5 km, northerly and southerly trade

winds are visible in the model and DWL profiles. A third feature, located to the south of the ITCZ and at around 10 km is the TEJ. While the position of the TEJ is well captured by the model, the DWL measurements suggest a model underestimation of the wind speed of around 5 ms$^{-1}$.

### 4.3 Dust plume in the Caribbean region

The synoptic situation presented in Figs. 6c and 7 shows a dust-loaded air mass reaching Barbados on 11 July 2013. This Saharan dust outbreak was the first to reach Barbados after the passage of the tropical storm Chantal over the Barbados region, between 8 and 9 July 2013. According to the results from the MACC model, this outbreak left the African continent 5 days before, on 6 July 2013, together with an AEW. In contrast to the 12 June and 17 June cases, the dust moved behind the AEW (AEW 3, Fig. 7) trough in this case and almost no dust was visible in front of the trough. This change is likely to

be due to the influence of the tropical storm Chantal and the associated wet deposition which can be clearly seen in Fig. 7 as a decrease in the AOD. On 11 July 2013, two research flights were performed. This case study focuses on the second flight, which departed from Barbados at 18:04 UTC and landed in Puerto Rico at 21:05 UTC. The DWL extinction and horizontal wind measurements are presented, together with the MACC model, in Fig. 10. A comparison between the DWL retrievals for the first flight on 11 July 2013 and the corresponding ground-based POLIS measurements is presented as part of the

DWL calibration validation in Chouza et al. (2015).

As in the previous cases, the modeled SAL shape and extinction is in relative good agreement with the DWL measurements, suggesting an adequate treatment of the Saharan dust long-range transport and deposition processes by the model. Both measurements and models show a dust plume with a descending upper bound and a rising lower bound as it moves westwards together with higher extinction coefficients in its lower half. The model results corresponding to the

measurements conducted close after the take-off from Barbados shows an aerosol plume extending from approximately 1.7 km up to 5 km bounded by two temperature inversions (not shown). This is compatible with the radio-sounding and POLIS ground-based lidar measurements presented in Groß et al. (2015), where the lidar measurements conducted on 11 July 2013 (23:00–24:00 UTC), five hour after take-off, show the SAL between 1.5 km and 4.8 km. In the same way as the model, the sounding temperature profiles exhibit inversions at its lower and upper bounds. In coincidence with previous measurements,

the ABL extinction is strongly underestimated by the model and a thin aerosol layer above the SAL is modelled, but not seen by the DWL.

As can be seen in Fig. 6c, the general circulation in the region is dominated by the Bermuda high pressure system, which leads to an anti-cyclonic flow over the Caribbean (Fig. 6c). The modelled winds are in relative good agreement with the measurements. The high wind speeds found between 4 km and 5 km are underestimated by the model by approximately 3

ms$^{-1}$, while its wedge shape and direction is well reproduced. The influence on the lee side of Dominica and Guadeloupe islands can be seen in the DWL retrievals for latitudes between 15° N and 16° N. These islands, with surfaces of 750 km$^2$ and 1600 km$^2$ and elevations of up to 1447 m and 1467 m respectively, introduce relative large disturbances in the low level

flow. This effect, characterized by a relative sudden decrease in the wind speed and a change in the direction for altitudes below 1.5 km cannot be captured by the MACC model due to its relative coarse grid of 80 km.

## 5 Summary and conclusions

Aerosol global models are of key importance not only for environmental and climate change studies, but also as air quality monitoring tools. The evaluation of the model capabilities by mean of comparisons with observations is of great importance for improvement of the models. In this study, the ability of the MACC model to reproduce the Saharan dust long-range transport across the Atlantic Ocean was investigated by means of a comparison with DWL, CALIOP and dropsonde observations conducted during SALTRACE. For first time a DWL was used to characterize the Saharan dust transport and its associated features.

First, the horizontal wind vector retrievals by the DLR airborne DWL during SALTRACE were evaluated with a comparison with collocated dropsonde measurements. The comparison shows a very good agreement, with a DWL-dropsonde mean difference of 0.08 ms$^{-1}$ and a standard deviation of 0.92 ms$^{-1}$ for the wind speed. For the case of the wind direction, the mean difference was 0.5° and the standard deviation 10°. These estimated accuracies are in agreement with previous studies (e.g. Weissmann et al., 2005).

The second part of this work focused on the evaluation of the MACC model using the wind and aerosol backscatter measurements from the DWL in combination with CALIOP extinction profiles and dropsondes. Two evaluation regions were defined: one close to the Saharan dust source at the West coast of Africa, and a second one in the Caribbean region. Although the wind comparison shows a general good agreement in both regions, a systematic underestimation of the AEJ wind speed was observed in the region close to West Africa. Since the AEJ is one of the main Saharan dust advection mechanisms, its underestimation would lead to a wrong estimation of the amount of dust being transported. Additionally, since the AEJ and the associated vertical and horizontal wind shear serve as energy source for the AEWs, the correct modelling of jet speed and position is crucial to correctly model the AWEs propagation and evolution. As indicated by Tompkins et al. (2005), a change in the aerosol radiative effect climatology can alter the AEJ speed. Thus, the observed underestimation in the AEJ wind speed could be partially due to the lack of radiative coupling between winds and aerosols in this MACC operational analysis run in 2013. Follow-on studies could explore this coupling of the aerosol and wind fields in a manner similar to Rémy et al. (2015), specifically for the SAL. Such a study would allow, in combination with the results presented in this work, to estimate the importance of an interactive radiative coupling to correctly simulate the AEJ intensity, the Saharan dust vertical distribution and its long-range transport.

Based on CALIOP extinction profiles, the model average aerosol vertical distribution was evaluated in both regions. Although a general good agreement was observed in the position and geometry of the SAL, a strong systematic underestimation of the marine boundary layer aerosol content was observed in both regions. Since the MACC model assimilates MODIS AOD measurements, the total model AOD is generally in good agreement with CALIOP measurements.

A slight overestimation of aerosol in the upper troposphere was observed in the model, which is likely to be an artifact introduced by the model to compensate the lack of aerosol in the marine boundary layer and thus match the assimilated MODIS AOD. An additional confirmation of this explanation could be investigated in future studies by means of a comparison between the model and in-situ aerosol measurements.

The good agreement between the modeled and measured AOD observed in this study serves as an indication of the potential of satellite data assimilation in global aerosol models. Currently, studies are being conducted in order to assimilate CALIOP attenuated backscatter vertical profiles in order to constrain the modeled aerosol vertical distribution. In a similar way, previous studies showed an improvement of modeled winds after the assimilation of airborne Doppler wind lidar measurements (Weissmann, et al., 2007). Future satellite missions like Aeolus (ESA, 2008) and EarthCARE (Illingworth, et

al., 2014) will provide a whole new set of wind and aerosol vertical profile measurements which are expected to lead to a significant improvement in weather prediction and global climate models, especially in regions where observations are sparse.

Additionally to the average wind and extinction comparison in both regions, three case studies were presented. These cases, which correspond to the initial, mid and final phase of the Saharan dust long-range transport process, allowed us to

investigate the characteristic features of the Saharan dust transport and the ability of the model to reproduce them. The DWL measurements carried out close to the West African coast show a SAL extending from 1 km up to 6 km and the AEJ at altitudes between 3 km and 6 km, with speeds of up to 25 ms$^{-1}$. Additionally, the pass of an AEW and the associated convective activity allowed us to investigate its effect on the dust distribution and transport pattern. As the dust is west transported over the Atlantic Ocean, the top of the SAL sinks, while its bottom rises. This typical SAL feature was

confirmed by the DWL retrievals corresponding to the second and third case studies. The measurements conducted over the Atlantic Ocean, corresponding to the second case study, showed a strong decrease in the dust load in the ITCZ as well as a rise in the lower edge of the SAL for low latitudes. The wind measurements carried out over the Atlantic revealed the presence of the AEJ north of the ITCZ, with speeds between 15 ms$^{-1}$ and 20 ms$^{-1}$ and altitudes between 2 km and 5 km. As the dust plume reached the Caribbean, approximately 5 days after leaving Africa, the top of the SAL was slightly below 5

km and its bottom at 2 km.

Although in all cases we found a good qualitative agreement between measurements and model, an underestimation of almost 10 m s$^{-1}$ in the AEJ speed was observed in the first case study. This is approximately two times the observed difference between the mean dropsonde measurements and the model. As expected, and due to the relative coarse resolution of the model, some island-induced mesoscale (<100 km) disturbances in the winds where observed by the DWL but not

reproduced by the model. The analysis of the extinction coefficient profiles shows similar results to those based on CALIOP and MACC spatio-temporal averages. The modelled extinction values in the boundary layer corresponding to the three case studies were far below the measured ones by the DWL, while above the SAL a thin aerosol layer was observed in the model but not in the DWL retrievals.

**Acknowledgements**

This work was funded by the Helmholtz Association under grant number VH-NG-606 (Helmholtz-Hochschul-Nachwuchsforschergruppe AerCARE). The SALTRACE campaign was mainly funded by the Helmholtz Association, DLR, LMU and TROPOS. CALIOP/CALIPSO data was obtained from the NASA Langley Research Center Atmospheric Science

5    Data Center. The SALTRACE flights on Cape Verde were funded through the DLR-internal project VolcATS (Volcanic ash impact on the Air Transport System). The first author thanks the German Academic Exchange Service (DAAD) for the financial support.

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

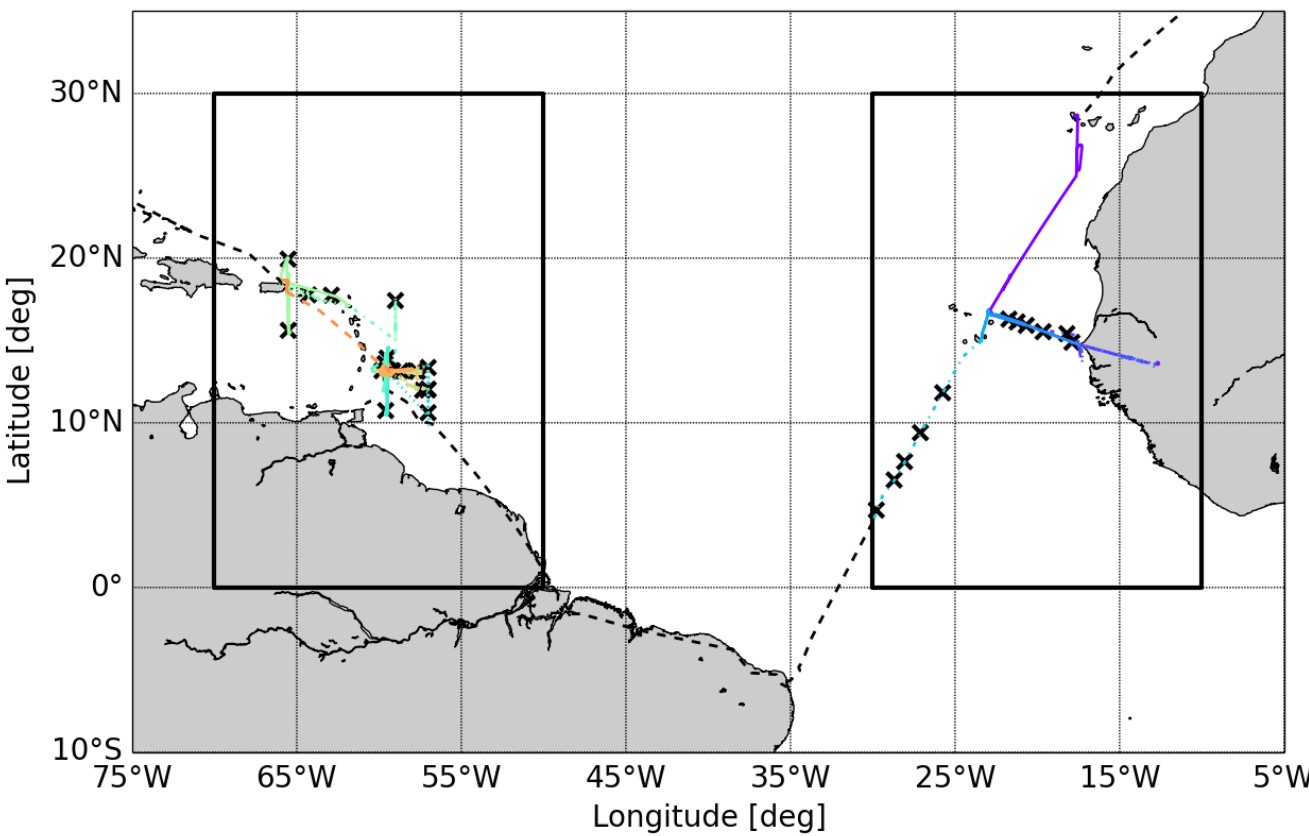

**Figure 1.** DLR Falcon flight tracks during SALTRACE. Dropsondes launch positions are indicated with black crosses. Black boxes indicate the regions which were used in this study (West Africa [0°-30°N; 10°-30°W] and Caribbean [0°-30°N; 50°-70°W]). Colored tracks indicate measurement flights included in this study. Black dashed tracks indicate transfer flights which are not used in this work.

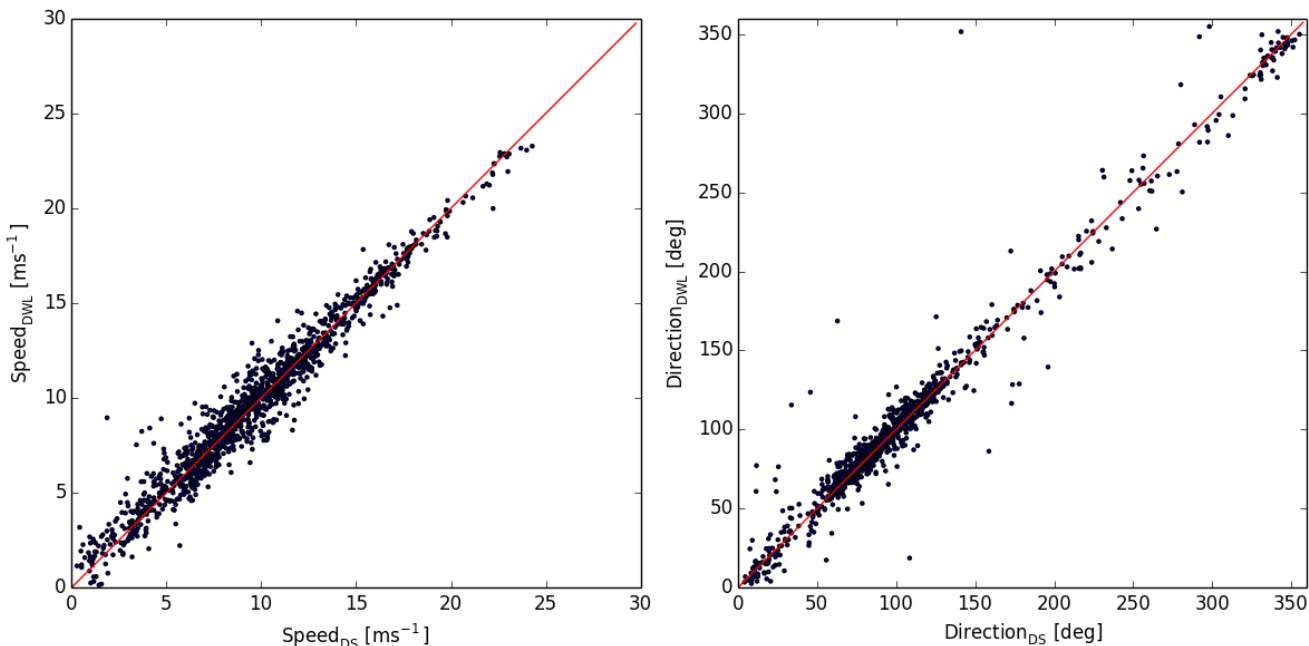

**Figure 2.** Comparison between dropsondes (DS) and DWL horizontal wind vector measurements. A total of 22 DWL/dropsondes profiles are included in this comparison, totalizing 1329 speed/direction pairs.

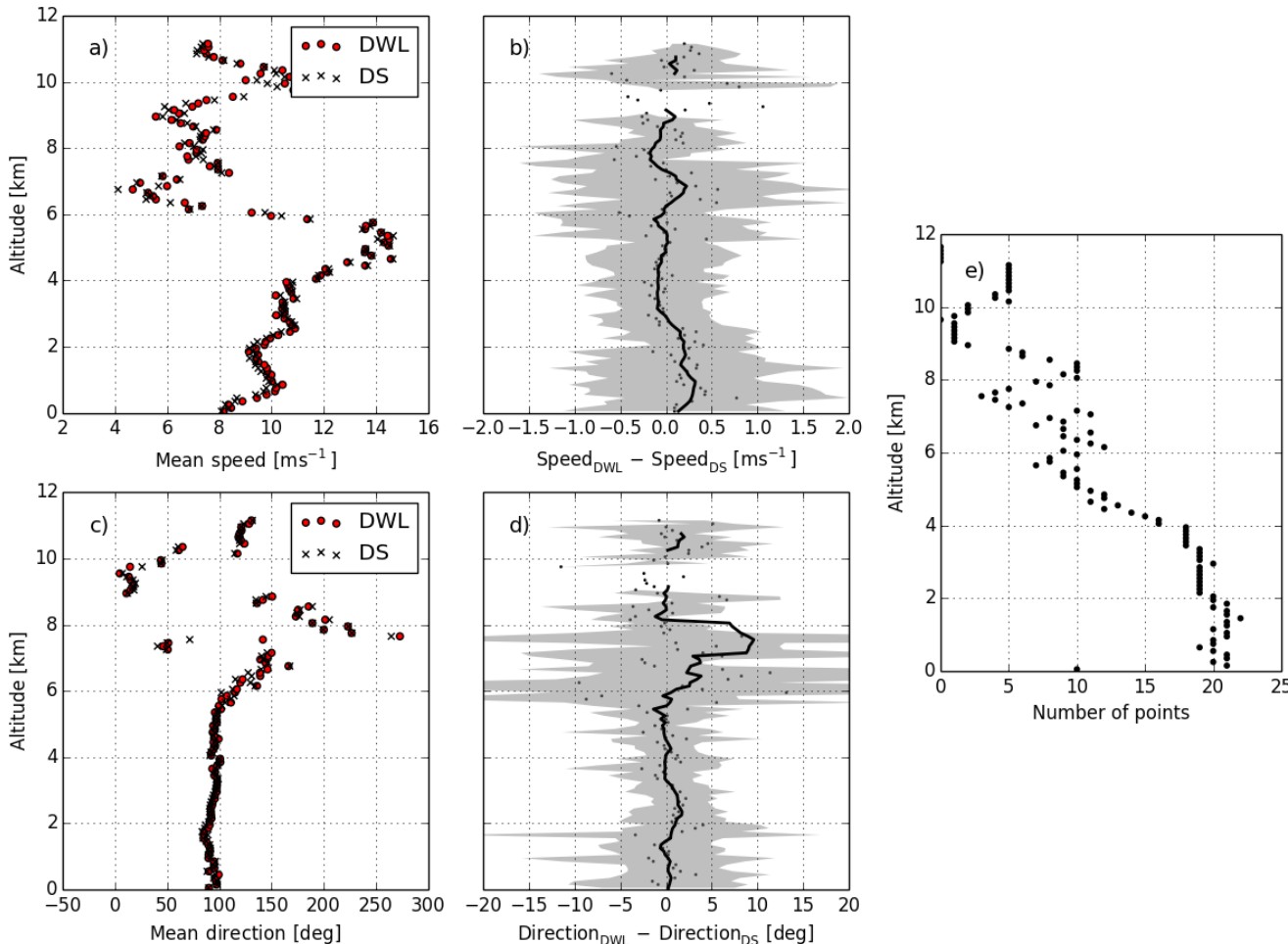

**Figure 3.** Comparison between 22 dropsondes (DS) and the corresponding collocated DWL horizontal wind vector measurements. a, c) Mean of the dropsonde (DS) and DWL wind speed and direction measurements, respectively. b, d) Difference between the mean DWL and DS wind speed and direction, respectively (dots, black), together with the corresponding 1000 m moving average (solid, black) and the standard deviation of the difference (shaded, grey). e) Number of compared measurement points as function of the altitude.

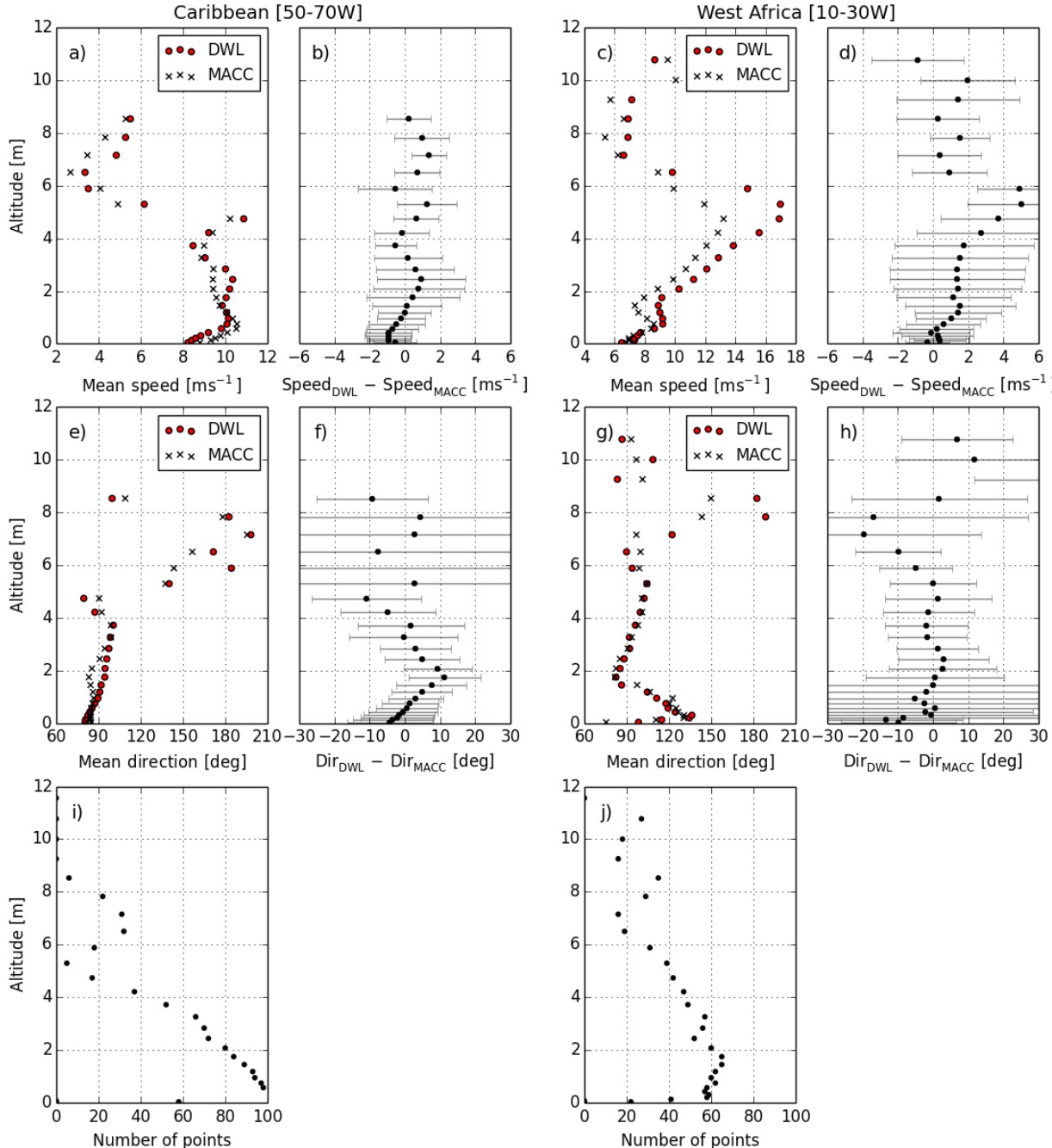

**Figure 4.** Comparison between the DWL and the MACC model horizontal wind measurements for the flights performed in the West African (right) and Caribbean (left) regions. a, c) Mean of the DWL and MACC wind speed measurements in the Caribbean and West African region, respectively. b, d) Difference between the mean DWL and MACC wind speed in the Caribbean and West African region, respectively (dots, black), together with the standard deviation of the difference (shaded,

grey). e, g) Same as a, c) but for the wind direction. f, h) Same as b, d) but for wind direction. i, j) Distribution of compared points as function of altitude in the West African (right) and Caribbean (left) regions, respectively.

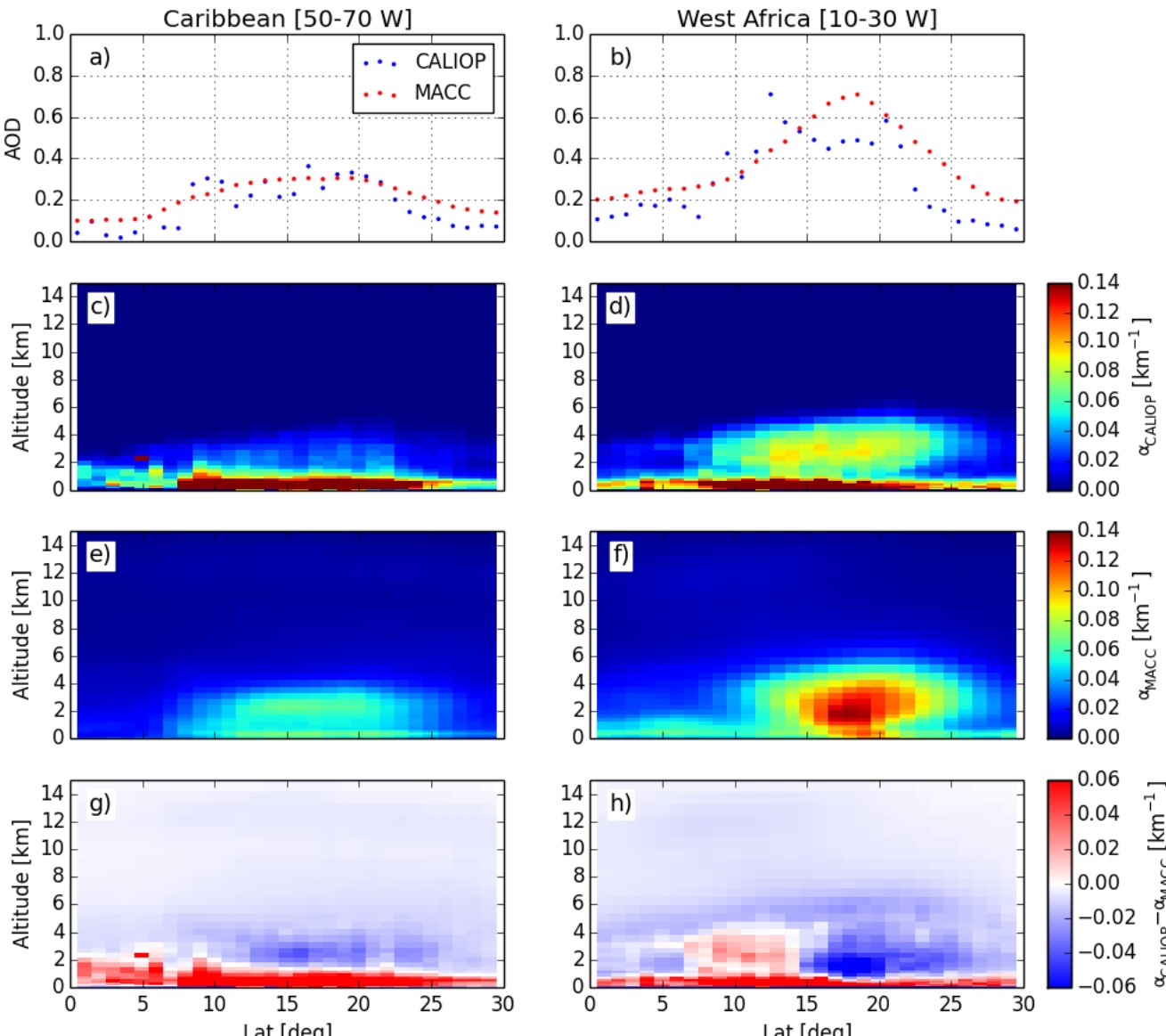

**Figure 5.** Comparison of CALIOP and MACC zonally averaged AOD and extinction coefficient for June-July 2013 in the regions of the study defined in Fig. 1. a, b) CALIOP and MACC AOD zonal mean. c, d) CALIOP extinction coefficient zonal mean. e, f) MACC extinction coefficient zonal mean. g, h) Difference between the zonally averaged extinction coefficient measured by CALIOP and simulated by MACC.

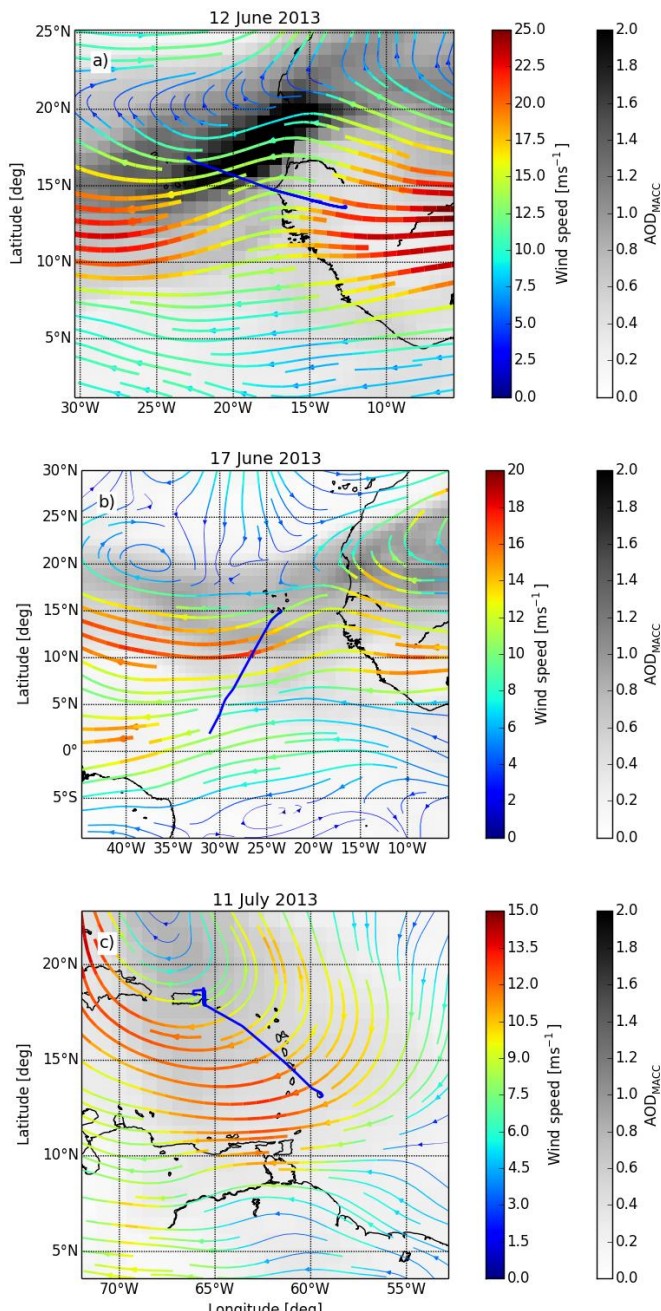

**Figure 6.** Horizontal winds at 700 hPa and AOD derived from the MACC model for the regions corresponding to the three cases: a) 12 June 2013 9:00 UTC, flight between Sal and Dakar. b) 17 June 2013 15:00 UTC: flight between Cape Verde and South America. c) 11 July 2013 18:00 UTC: flight between Barbados and Puerto Rico. The sections of the flight tracks shown in Figs. 8, 9 and 11 (blue, solid), wind streamlines (in m s$^{-1}$) and AOD (grey color scale) are shown.

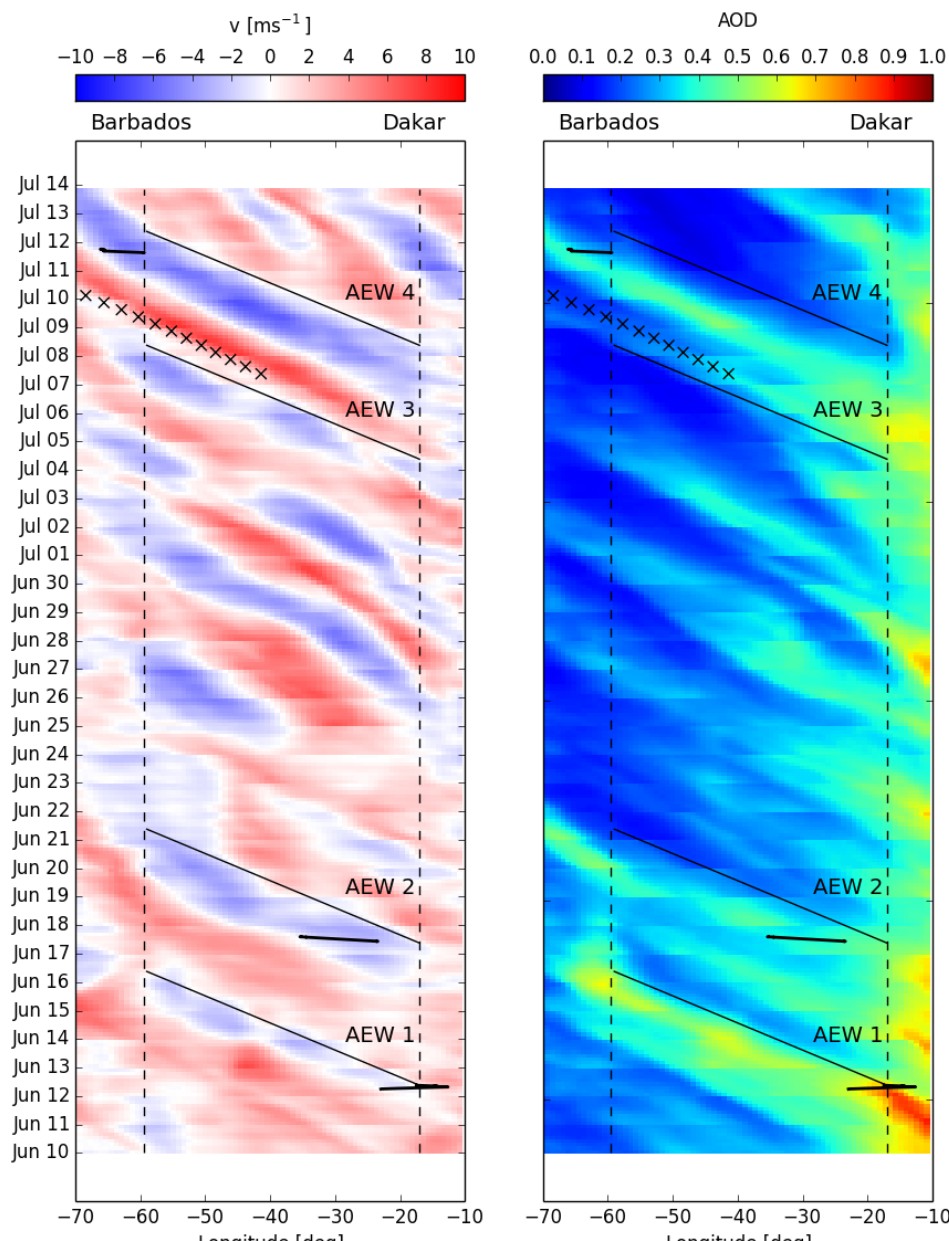

**Figure 7.** MACC model meridional wind (left) and AOD (right) presented in form of a Hovmöller diagram averaged for latitudes between 0°N and 30°N and 700 hPa pressure level. The flights corresponding to each case (black, thick, solid), the longitude of Dakar and Barbados (black, dashed), relevant AEWs troughs (black, solid) and the position of the tropical storm Chantal (black, crosses) are indicated.

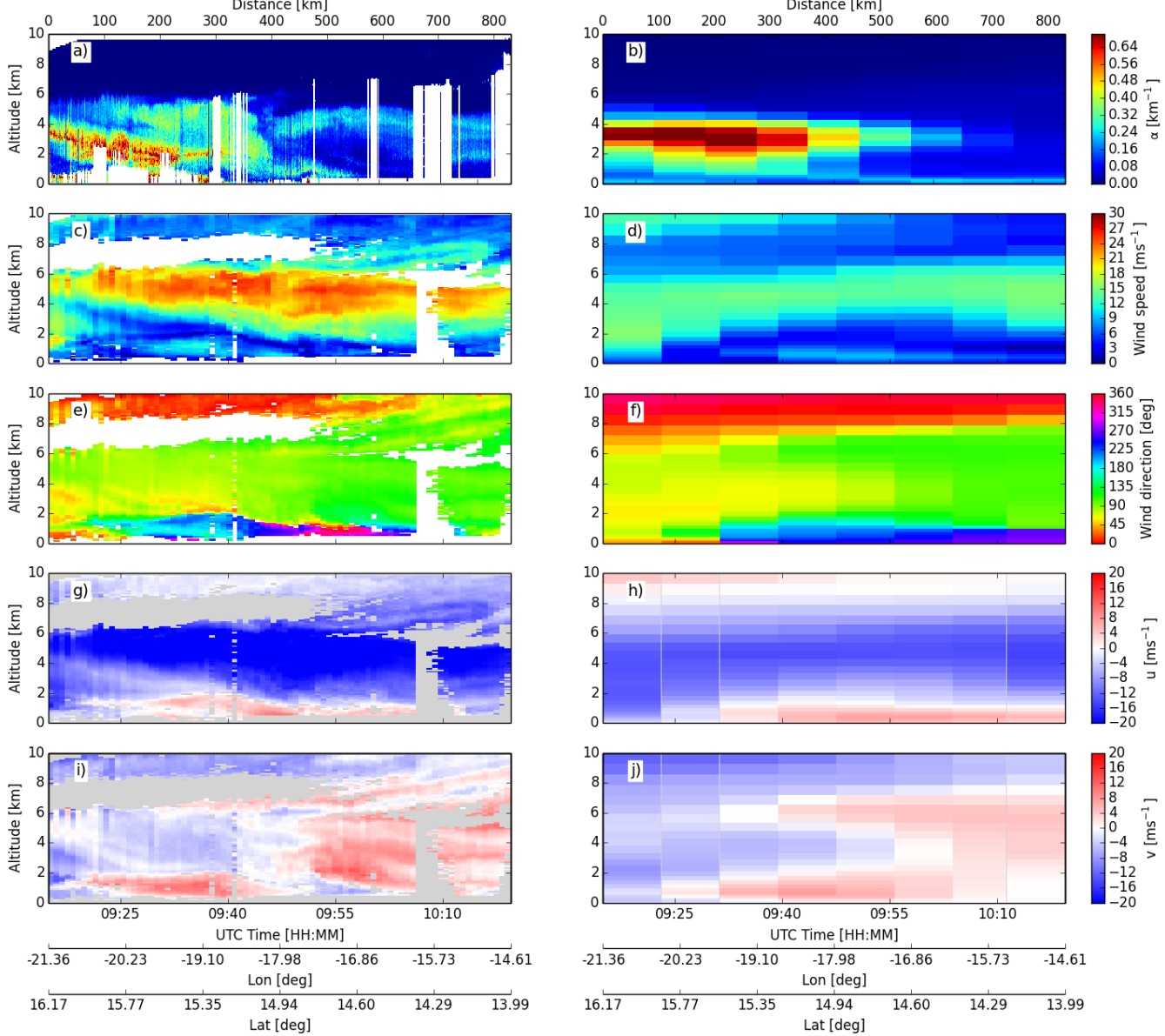

**Figure 8.** DWL measurements (left column) and MACC model (right column) along the measurement track for the flight on 12 June 2013. a, b) Extinction coefficient. c, d) Horizontal wind speed. e, f) Horizontal wind direction. g, h) Zonal wind component. i, j) Meridional wind component. Regions were no atmospheric signal is available (e.g. below clouds, low laser energy, low aerosol load) are colored white in panels a, c, e and grey in panels g, i.

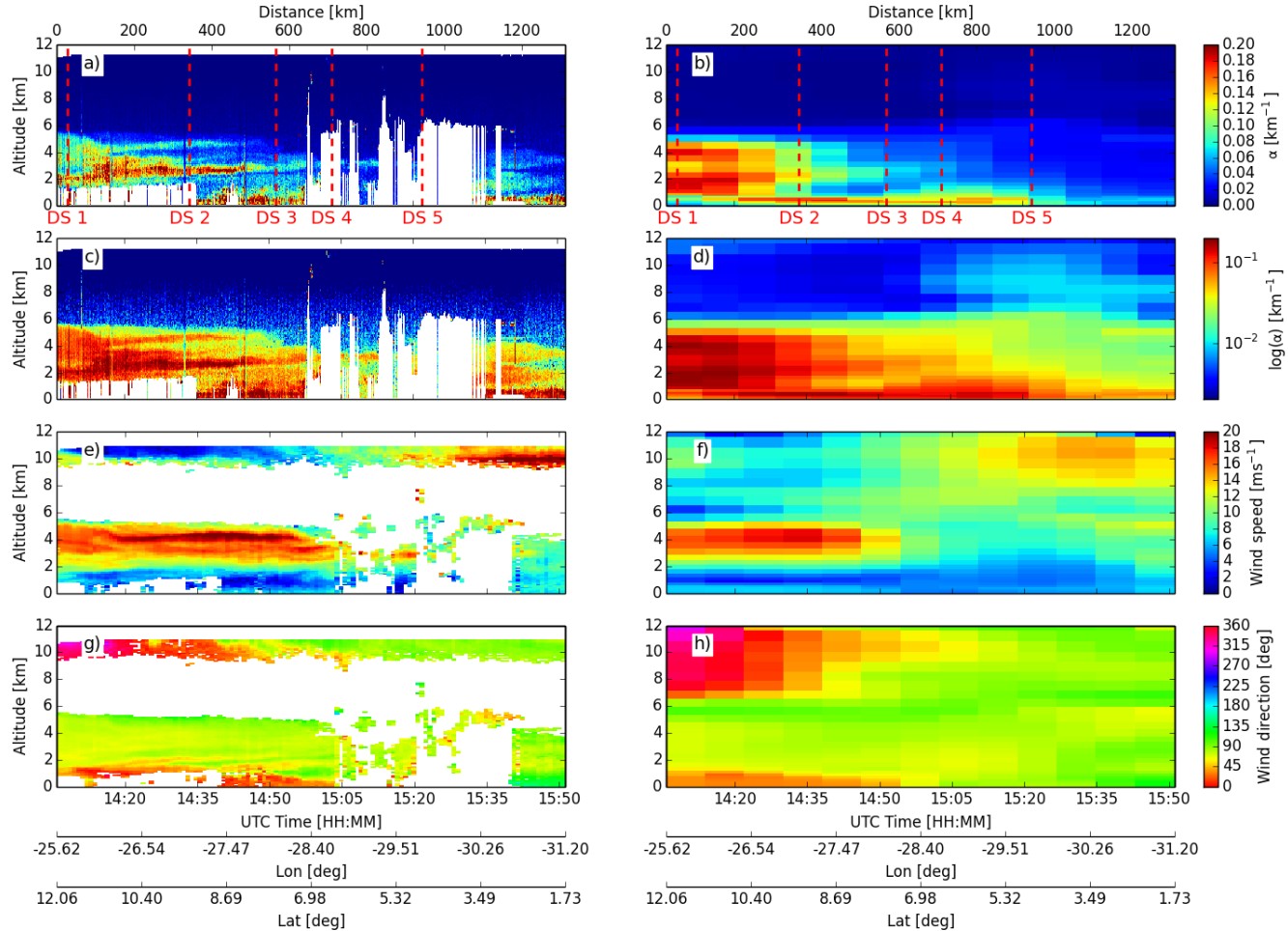

**Figure 9.** DWL measurements (left column) and MACC model (right column) along the measurement track for the flight on 17 June 2013. a, b) Extinction coefficient. c, d) Extinction coefficient plotted in logarithmic scale. e, f) Horizontal wind speed. g, h) Horizontal wind direction. The white color indicates regions were no atmospheric signal is available (e.g. below clouds, low laser energy, low aerosol load). The dropsondes launch time (dashed, red) is indicated in a) and b).

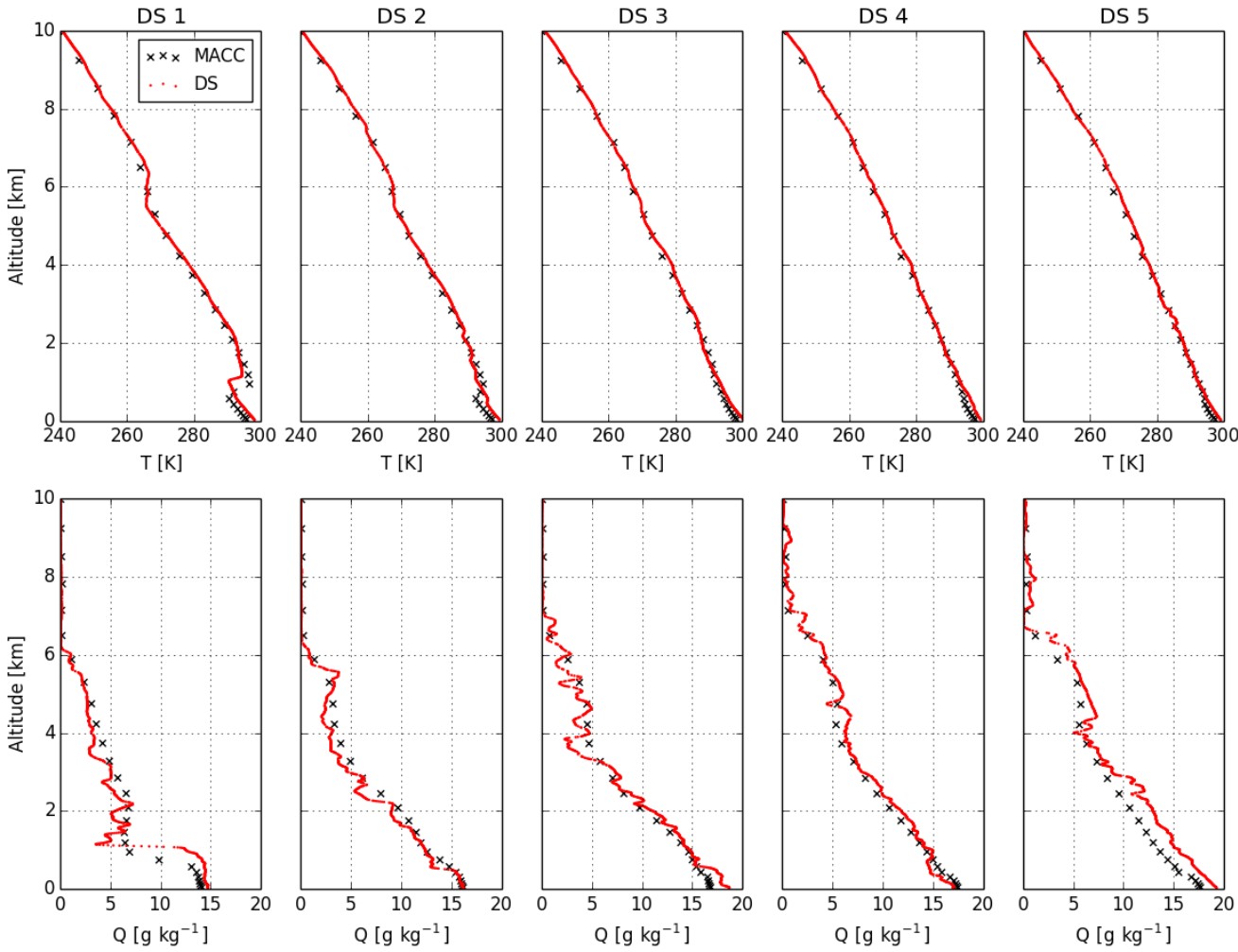

**Figure 10.** Temperature (upper row) and water vapor mixing ratio (lower row) measured by the dropsondes launched during the flight on 17 June 2013 (dots, red) and the corresponding MACC model values (crosses, black).

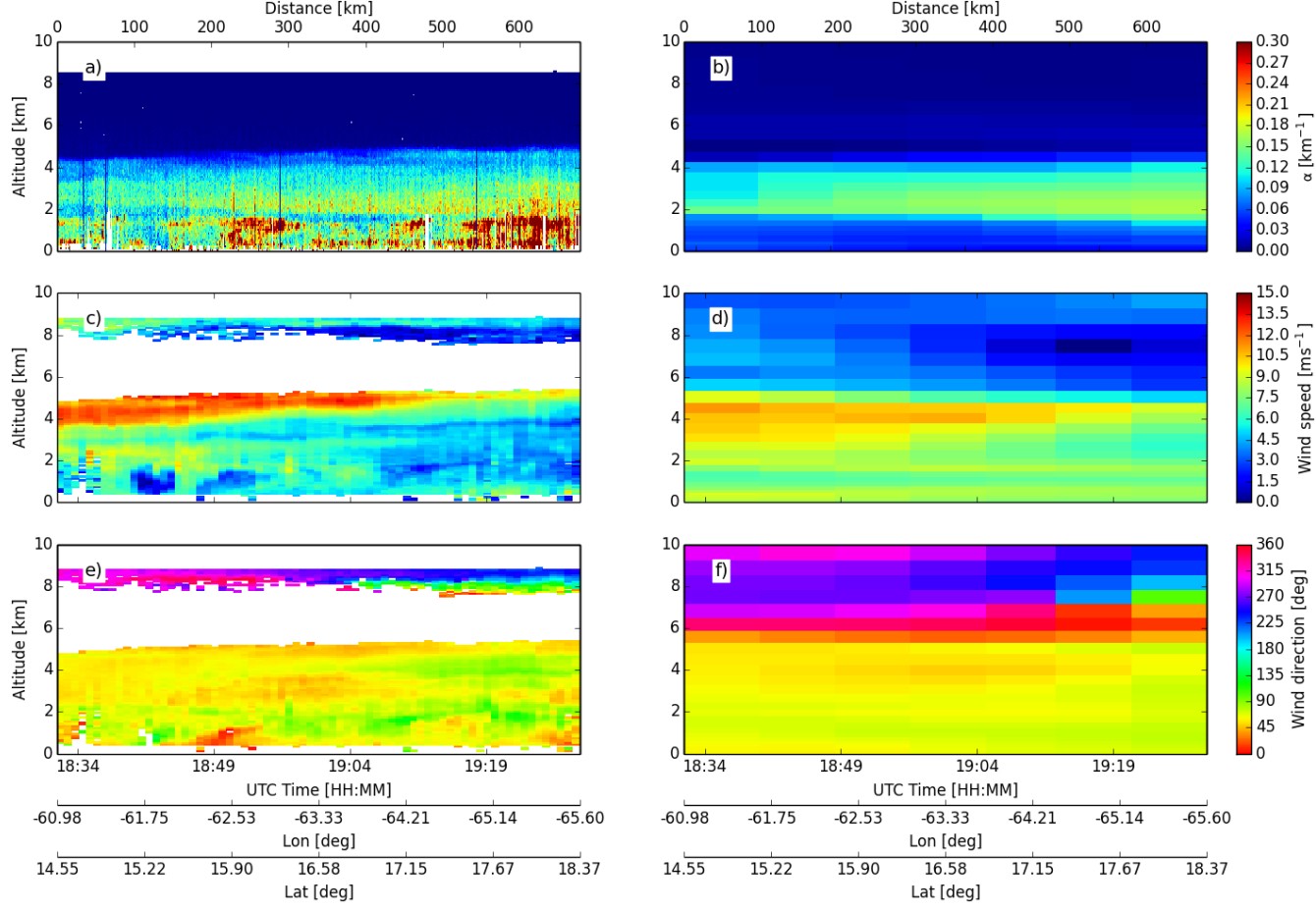

**Figure 11.** Same as Fig. 8, but for the flight on 11 July 2013. Zonal and meridional wind components are omitted.