# Peer review of "Saharan dust long-range transport across the Atlantic studied by an airborne Doppler wind lidar and the MACC model"

_Atmospheric Chemistry and Physics, 2016_

## Referee Comment (RC1) · Anonymous Referee #1 · 27 Jun 2016

The authors use airborne measurements from the SALTRACE campaign, CALIPSO products and the MACC model simulations to describe case studies of Saharan dust long-range transport over the Atlantic. This is a work of very good quality where sophisticated data/models are used for the description of the SAL during an experimental campaign of high importance.

However, the focus of the paper is mostly on the MACC model evaluation, which puts a large portion of the work shown out of the scope of ACP. I think that the paper needs major revisions for being ACP-compatible and this can be achieved with a more in-depth analysis of the dust-related physical processes revealed during the experiment. In short, I propose to not just compare the model output with observations but dis-

[Figure]

Interactive
comment

cuss also the physical meaning of these results. This can be done in section 4 (case studies).

Beyond this major revision, I have only the following minor comments (sorted by importance):

Section 3.3: Aerosol extinction: Here I would urge the authors to revisit the related literature concerning the CALIPSO extinction underestimation for the dust case (see for example Tesche et al., JGR; Wandinger et al. in GRL; Amiridis et al. in ACP). It is possible that MACC performs better in terms of AOD than CALIPSO for the cases mentioned in the paper, especially because MODIS is assimilated over ocean.

Abstract: The authors mention: DWL measurements are validated against dropsondes. This task is not to be mentioned in the abstract (and out of the scope of ACP). Even though the evaluation exercise provides confidence on DWL retrievals, it doesn't have to be mentioned here.

Page 5, Line 22: Please revise the web link, you provide the site for CERES and not for CALIPSO.

Page 12, line26: It is Figure 9 I think (instead of 7)

Figure 8: Please use a different than white color for the non-available data in the left panel in order to distinguish from the right panel, where white stands for the zero values.

Too much info on MACC in Section 2. I think that appropriate references exist in the literature, so I would avoid so detailed description.

There are many typos throughout the document, please give it a thorough read and revise accordingly (e.g. be careful with the use of "where" instead of "were", a mistake repeated many times).

---

## Referee Comment (RC2) · Anonymous Referee #2 · 14 Jul 2016

General:

The paper is based on airborne Doppler lidar observations collected during a unique and complex field campaign in the Caribbean. The paper is appropriate for the Special issue of SALTRACE. However, the paper needs major revisions.

I have problems with all the aerosol-related results and comparisons. Spaceborne lidar observations, coherent lidar observations and MACC aerosol transport model computations are compared. I do not fully trust the CALIOP observations (my reasons are given below), I do not fully trust the Doppler lidar estimates of aerosol extinction, so that all the comparisons and conclusions of the paper do not really convince me. So, at the end, I do not know whether the MACC aerosol products or the lidar products

show the truth.

Especially, I am missing the integration of the ground-based SALTRACE aerosol Raman lidar observations into the study. They deliver the most trustworthy aerosol profiles. To convince the reader, in the first place, the Doppler lidar data of particle backscattering and extinction estimates should be compared with Raman lidar observations for some Barbados cases. After demonstrating the usefulness of the Doppler lidar for quantitative aerosol profiling in the Barbados area..., one may continue with comparisons with CALIOP products. So, my main point is that I am missing SALTRACE ground-based with airborne lidar comparison of aerosol profiles.

Please keep in mind, your paper is a contribution to a SALTRACE Special Issue. So the reader expects complex papers with complex integration of lidar and model results.

Details:

Page 2, line 7: I strongly recommend to check the special issues of SAMUM 1 and 2 for potential references. For example, why did you (the SAMUM-SALTRACE science group) make so many observations at Cabo Verde on SAL (winter campaign as well the SAMUM-2 lidar campaign in summer 2008) and finally there is no SAMUM reference at all. This is not professional, I mean other groups usually put all the own papers in the foreground whereas you seem to ignore them simply. I have to assume that you do not know the SAMUM special issues.

Page 2: You need to explain all abbreviations when they appear the first time. ALL! SAMUM, AMMA, NAMMA, BERTHA, POLIS, CALIPSO etc.

Page 3: There are references for SALTRACE station at CIMH (Gross 2015, 2016, Haarig ILRC 27 New York, Toledano ?), should be given....

Page 5, before the CALIOP section: I think we need a small section on the retrieval of the 532 nm backscatter coefficient from the Doppler lidar observations. This not trivial. Please provide something like a summary (step by step of the entire retrieval scheme)

and provide uncertainties.

Page 5, line 18: CALIOP provides profiles of the particle backscatter coefficient, and then with the help of lidar ratios the extinction and AOD is estimated. The forward Klett algorithm has to be applied. As a consequence, the solution profiles are rather uncertain, and they are rather sensitive to uncertainties in the assumed lidar ratio profile (input profile). That should be made very clear. This approach is rather different from the SALTRACE Raman lidar approach. Did you check the lidar ratios used by CALIOP and the ones obtained by the SALTRACE lidars. To my knowledge, CALIOP uses 40 sr for dust extinction conversion. However, dust lidar ratios are typically close to 60sr….So, there may be an error (systematic bias of 30%), and because of the fact that the solutions get more and more unstable from the tropopause towards the ground, the full profile gets increasingly corrupted with range.... towards the surface. Please check and comment on that.

Page 5, line 20, dust vertical distribution of what??? The vertical distribution is not a parameter!

Page5, lines 22-28: Are their CALIOP overflights over Barbados! . . . so that one could check the quality of the CALIOP profiles directly with Raman lidar profiles.

Page 5, line 32: Explain abbreviation MACC

Page 8, line 16: Extinction coefficient and AOD are reported….., so no separation of dust and non-dust particle extinction? Just the total particle extinction! What lidar ratios are then used….., for the dust/marine mixture from about 1000-1500 m (SAL base) down to 400 m (below 400 m one may assume pure marine…)

Section 3.3. It is a bit confusing, when just total particle extinction coefficients are compared without an approach to separate dust from marine….

Page 10, line 2: Do you believe that ext coef. obtained from CALIOP is just 50 Mm-1 plus minus 20Mm-1 in the ABL, and all this is just marine aerosol…?

Page 10, line 10: Who is right? MACC may be wrong, such layers reaching to 20 km height are not 'realistic'. And CALIOP cannot resolve such tiny aerosol traces.

Page 11, line18-20: I have many questions regarding the comparisons, without having a good answer, how to handle the quality of the comparisons. I would like to see a very careful discussion. One should again mention all the potential errors sources, and clearly state that the comparisons are combined with high uncertainties. As you know, CALIOP delivers only particle backscatter profiles. The Klett forward integration method is used. The solution can thus be very erroneous, especially at the end of the profile (lowest part of the atmosphere, i.e., in your cases, in the lowest part of the SAL and the layer below the SAL down to the ground). The uncertainty can easily be 30-50%, even larger. On the other hand, your DWL does not allow to retrieve rather accurate backscatter coefficients, too. The conversion to 532nm backscatter is combined with high uncertainties, and the further conversion to 532 nm extinction, as well. This introduces a systematic bias to the entire profile within a given layer (SAL, transition layer, marine MBL), and this bias is different for the different layers. How can you then state, for example, that MACC underestimates the extinction of the marine boundary layer and overestimates the extinction of the SAL? So I would like to see a very sensitive discussion in view of all the uncertainties on both sides, observations and modelling. . ..

Page 12, lines 23-25, so if CALIOP cannot measure the 'artifact' (as produced by the MACC model) why do you not at least check the SALTRACE ground-lidar data, whether there was an aerosol layer in the upper free troposphere or not. Such layers at such great heights are clearly a large scale phenomenon. . ... and should have been seen by the Barbados lidars, if existing.

Page 13, section 4.3. This could be the central subsection of the entire paper. Here one could start with the comparison of DWL backscatter (and extinction profiles) with ones from the ground lidars. Afterwards, one could step forward with CALIOP and model output discussions.

The DWL extinction profiles... what lidar ratios did you use? For the dust layer, for the mixed dust/marin layer, for the marine layer....? Is that in agreement with ground lidar data ? The ground-based lidars measure lidar ratios during darkness, and these lidar ratios are certainly valid hours before or later..., and thus applicable to the DWL observations.

At the end I must say: It is quite strange to see an aerosol-related comparison paper on the basis of airborne Doppler lidar measurements, a lidar which does not measure backscatter coefficients...., furthermore based on CALIOP which does not measure extinction profiles, and the only lidars, delivering extinction profiles are not included in the paper

I found a SAMUM 2 Raman lidar vs CALIOP intercomparison paper, Teshe et al. (JGR, 2013). You may not know that paper, but it should be referenced.... More general, one should check and know all the SAMUM papers from 2009 and 2011 and provide proper referencing to all the SAMUM efforts done.

Some comments to the figures:

Do we need Figure 2 in this paper on CALIOP and MACC? Is that not already presented in wind-related SALTRACE papers?

Figure 3: Same question...

Figure 5: Why do you show this figure? You show two-month mean values, right? From all the CALIOP observations in June and July 2013? For proper comparison, the respective MACC results were averaged for the same CALIOP observational times within the two months?

Figures 5 a and b: Who is right? CALIOP or MACC? Who knows, I do not know? Because MACC is based on MODIS AOD, I would trust MACC. Because CALIOP needs lidar ratios they do not measure, and thus do not know..... these AOD values are less trustworthy. What lidar ratio did they use for dust 40sr or 55sr? CALIOP suffers from

[Figure]

multiple scattering effects in dust. Is that taken into account. MS leads to underestimation of AOD.

Figures 5 g and h.... Again, because the forward integration Klett method has to be used (very sensitive to uncertainties in the lidar ratio input profile), I do not trust the CALIOP values in the MBL and the mixing zone above MBL.... Again, who is right? There is no answer! Should be critically discussed! The only answer could be given by the SALTRACE ground lidars....

So, at the end, Figure 5 shows a comparison of MACC vs CALIOP! No DWL observation at all.

Figure 8: Again! Who is right (in figs 8 a and b). DWL cannot measure extinction. This is a wrong statement. The lidar can even not measure backscatter! It needs help by 'real'aerosol lidars. So I am always puzzled, what the basic and essential goal of this paper is...? Yes, the wind data comparison is very attracting, very convincing! This is the strongest part of the paper. MACC obviously does a good job.

Figure 9: Again......, the only reliable information (in a and c) is the observation of dust layering...

Figure 10: Is this figure needed. Ok, convincing! MACC does a good job! If that is an important finding, leave it in. If not, remove the figure.

Figure 11: At least for 11 July, I expected to see a broader view on the aerosol situation (conditions). Here, I would like to see the other 'real' SALTRACE aerosol lidar observations... in comparison with the DWL observations.

I am always confused by the fact, that this will be a contribution to a SALTRACE Special Issue, but the special issue aspect, integration of all available measurements to design a complete aerosol picture, is only poorly given. It seems to me that authors need publications and do not really take care and the time to look at all available data.

---

## Author Comment (AC1) · 8 Aug 2016

Dear reviewers,

We like to thank you for your helpful comments on our paper titled "Saharan dust long-range transport across the Atlantic studied by an airborne Doppler lidar and the MACC model".

The original comments are in bold, followed by our replies.

**Reviewer #1**

**General comments:**

**The authors use airborne measurements from the SALTRACE campaign, CALIPSO products and the MACC model simulations to describe case studies of Saharan dust long-range transport over the Atlantic. This is a work of very good quality where sophisticated data/models are used for the description of the SAL during an experimental campaign of high importance.**

**However, the focus of the paper is mostly on the MACC model evaluation, which puts a large portion of the work shown out of the scope of ACP. I think that the paper needs major revisions for being ACP-compatible and this can be achieved with a more indepth analysis of the dust-related physical processes revealed during the experiment.**

**In short, I propose to not just compare the model output with observations but discuss also the physical meaning of these results. This can be done in section 4 (case studies).**

In our view, and based on previous papers published on ACP, we consider that a publication evaluating the current modelling capabilities based on model-measurement comparisons is within the scope of ACP. As examples of such kind of studies we can cite:

- Cuevas, E., Camino, C., Benedetti, A., Basart, S., Terradellas, E., Baldasano, J. M., Morcrette, J. J., Marticorena, B., Goloub, P., Mortier, A., Berjón, A., Hernández, Y., Gil-Ojeda, M., and Schulz, M.: The MACC-II 2007–2008 reanalysis: atmospheric dust evaluation and characterization over northern Africa and the Middle East, Atmos. Chem. Phys., 15, 3991-4024, doi:10.5194/acp-15-3991-2015, 2015.

- Wagner, A., Blechschmidt, A.-M., Bouarar, I., Brunke, E.-G., Clerbaux, C., Cupeiro, M., Cristofanelli, P., Eskes, H., Flemming, J., Flentje, H., George, M., Gilge, S., Hilboll, A., Inness, A., Kapsomenakis, J., Richter, A., Ries, L., Spangl, W., Stein, O., Weller, R., and Zerefos, C.: Evaluation of the MACC operational forecast system – potential and challenges of global near-real-time modelling with respect to reactive gases in the troposphere, Atmos. Chem. Phys., 15, 14005-14030, doi:10.5194/acp-15-14005-2015, 2015.

- Kunz, A., Spelten, N., Konopka, P., Müller, R., Forbes, R. M., and Wernli, H.: Comparison of Fast In situ Stratospheric Hygrometer (FISH) measurements of water vapor in the upper troposphere and lower stratosphere (UTLS) with ECMWF (re)analysis data, Atmos. Chem. Phys., 14, 10803-10822, doi:10.5194/acp-14-10803-2014, 2014.

- Cesnulyte, V., Lindfors, A. V., Pitkänen, M. R. A., Lehtinen, K. E. J., Morcrette, J.-J., and Arola, A.: Comparing ECMWF AOD with AERONET observations at visible and UV wavelengths, Atmos. Chem. Phys., 14, 593-608, doi:10.5194/acp-14-593-2014, 2014.

Also the second reviewer states that this work is suitable for the ACP SALTRACE special issue. Nevertheless, and as was suggested by the reviewer, we will expand the discussion on the modelled and observed physical processes in the revised manuscript.

**Minor comments:**

**Section 3.3: Aerosol extinction: Here I would urge the authors to revisit the related literature concerning the CALIPSO extinction underestimation for the dust case (see for example Tesche et al., JGR; Wandinger et al. in GRL; Amiridis et al. in ACP). It is possible that MACC performs better in terms of AOD than CALIPSO for the cases mentioned in the paper, especially because MODIS is assimilated over ocean.**

A description of the uncertainties associated by the CALIOP retrievals will be included in the revised manuscript together with the proposed references from Tesche et al. and Wandinger et al.. A detailed discussion of this issue can be found in the answer to the second reviewer.

**Abstract: The authors mention: DWL measurements are validated against dropsondes. This task is not to be mentioned in the abstract (and out of the scope of ACP). Even though the evaluation exercise provides confidence on DWL retrievals, it doesn't have to be mentioned here.**

The results of the DWL measurement validations of the horizontal wind will be removed from the abstract in the revised version. We will keep the results in Section 2.2 because it provides very good confidence in the DWL wind measurements, which in turn form the basis of the MACC evaluation.

**Page 5, Line 22: Please revise the web link, you provide the site for CERES and not for CALIPSO.**

Right, this will be corrected.

**Page 12, line26: It is Figure 9 I think (instead of 7)**

Right, this will be corrected.

**Figure 8: Please use a different than white color for the non-available data in the left panel in order to distinguish from the right panel, where white stands for the zero values.**

The figure will be modified in the revised manuscript to take in account the difference non-available data and zero values.

**Too much info on MACC in Section 2. I think that appropriate references exist in the literature, so I would avoid so detailed description.**

We have included relevant information about MACC regarding aerosol modelling and assimilation. This is needed for readers which are not familiar with MACC and for the understanding of the subsequent analysis. The model description will be slightly shortened in the revised version of the manuscript.

**There are many typos throughout the document, please give it a thorough read and revise accordingly (e.g. be careful with the use of "where" instead of "were", a mistake repeated many times).**

We will read the manuscript thoroughly and correct it accordingly.

**Reviewer #2**

**General comments:**

**The paper is based on airborne Doppler lidar observations collected during a unique and complex field campaign in the Caribbean. The paper is appropriate for the Special issue of SALTRACE. However, the paper needs major revisions.**

**I have problems with all the aerosol-related results and comparisons. Spaceborne lidar observations, coherent lidar observations and MACC aerosol transport model computations are compared. I do not fully trust the CALIOP observations (my reasons are given below), I do not fully trust the Doppler lidar estimates of aerosol extinction, so that all the comparisons and conclusions of the paper do not really convince me. So, at the end, I do not know whether the MACC aerosol products or the lidar products show the truth.**

**Especially, I am missing the integration of the ground-based SALTRACE aerosol Raman lidar observations into the study. They deliver the most trustworthy aerosol profiles. To convince the reader, in the first place, the Doppler lidar data of particle backscattering and extinction estimates should be compared with Raman lidar observations for some Barbados cases. After demonstrating the usefulness of the Doppler lidar for quantitative aerosol profiling in the Barbados area…, one may continue with comparisons with CALIOP products. So, my main point is that I am missing SALTRACE ground-based with airborne lidar comparison of aerosol profiles.**

**Please keep in mind, your paper is a contribution to a SALTRACE Special Issue. So the reader expects complex papers with complex integration of lidar and model results.**

Although we agree with the reviewer regarding the accuracy of the extinction retrievals provided by CALIOP and the DWL, and the need of discussing it in the paper. Their retrievals are still good enough to sustain the claims presented in this work. Whether or not a retrieval is useful depends on the conclusion intended to be proven.

Additionally, it has to be noted that this paper doesn't intend to present accurate measurements of the aerosol optical properties, but a comparison of the modelled and measured dynamics and patterns involved in the Saharan dust long-range transport process (e.g. shape and position of the SAL, position and strength of the African

Easterly Jet, modulation of the dust transport by the African Easterly Waves, etc.). For this purpose, the aerosol two-dimensional spatial distribution measurements provided by CALIOP and by the airborne DWL provide more adequate results than the vertical profiles of ground-based aerosol lidars at one location. This is especially true for the DWL, where together with an estimation of the extinction coefficient, accurate wind measurements can be retrieved. In our view, the reviewer is misinterpreting the purpose of a study which doesn't intend to provide an accurate characterization of the aerosol optical properties but provide an overview picture of the Saharan dust long-range transport process and the current modelling capabilities.

Finally, the major concern of the reviewer regarding the lack of an evaluation of the DWL retrieval used in this work is answered by an AMT paper published in this special issue: Chouza et al., 2015. This publication presents an overview of the retrieval method and an estimation of it accuracy by means of a comparison with the ground-based aerosol POLIS in the Barbados region and with CALIOP in the West African coast region.

**Page 2, line 7: I strongly recommend to check the special issues of SAMUM 1 and 2 for potential references. For example, why did you (the SAMUM-SALTRACE science group) make so many observations at Cabo Verde on SAL (winter campaign as well the SAMUM-2 lidar campaign in summer 2008) and finally there is no SAMUM reference at all. This is not professional, I mean other groups usually put all the own papers in the foreground whereas you seem to ignore them simply. I have to assume that you do not know the SAMUM special issues.**

We certainly know the SAMUM special issues. Adequate references (e.g. Tesche et al.) will be included and discussed in the revised version of the manuscript.

**Page 2: You need to explain all abbreviations when they appear the first time. ALL! SAMUM, AMMA, NAMMA, BERTHA, POLIS, CALIPSO etc.**

An adequate explanation of each abbreviation will be added in the revised version of the manuscript.

**Page 3: There are references for SALTRACE station at CIMH (Gross 2015, 2016,Haarig ILRC 27 New York, Toledano ?), should be given.**

References for the SALTRACE station at CIMH will be added in the revised version of the manuscript.

**Page 5, before the CALIOP section: I think we need a small section on the retrieval of the 532 nm backscatter coefficient from the Doppler lidar observations. This not trivial. Please provide something like a summary (step by step of the entire retrieval scheme).**

A reference to the retrieval scheme (Chouza et al., 2015) is included on P. 4, line 24. This publication presents not only the calibration procedure, but also an answer to one of your major concerns, the accuracy of the DWL retrievals. An estimation of the retrieval accuracy by means of a comparison with the ground-based aerosol POLIS in the Barbados region and with CALIOP in the West African coast region is presented in that paper.

A summary of the comparison results and accuracy estimation will be added to the revised version of this manuscript.

**Page 5, line 18: CALIOP provides profiles of the particle backscatter coefficient, and then with the help of lidar ratios the extinction and AOD is estimated. The forward Klett algorithm has to be applied. As a consequence, the solution profiles are rather uncertain, and they are rather sensitive to uncertainties in the assumed lidar ratio profile (input profile). That should be made very clear. This approach is rather different from the SALTRACE Raman lidar approach. Did you check the lidar ratios used by CALIOP and the ones obtained by the SALTRACE lidars. To my knowledge, CALIOP uses 40 sr for dust extinction conversion. However, dust lidar ratios are typically close to 60sr. So, there may be an error (systematic bias of 30%), and because of the fact that the solutions get more and more unstable from the tropopause towards the ground, the full profile gets increasingly corrupted with range.... towards the surface. Please check and comment on that.**

A clarification on the uncertainties associated by the CALIOP measurements and extinction estimation (including lidar ratios) will be added to the revised version of the manuscript.

An underestimation by 30% in the LR would certainly lead to an underestimation of the extinction coefficient in the boundary layer. Even considering this underestimation, MACC shows values between 50% and 100% **lower** than CALIOP. The general conclusion about the underestimation of the MBL by MACC is not affected by the underestimation from CALIOP.

On the other hand, the results indicate an overestimation on the average SAL extinction. The following figure shows the meridional average of the extinction presented in Fig. 5 (left plot corresponds to the Caribbean region, while the right plot corresponds to the West African region), together with the relative difference between CALIOP and MACC and the mean lidar ratio used for the CALIOP inversion. Even if the extinction on the SAL is multiplied by a correction factor to take in account the underestimation of the lidar ratio by CALIOP (55sr/40sr), most of the MACC results will be still above the CALIOP retrievals. Certainly the difference will be lower, but the SAL AOD derived from MACC is still larger than the AOD derived from CALIOP extinction. Based on SAL lower and upper bounds between 2 and 5 km in the Caribbean and between 1 and 6 km in the African region, the SAL AOD values based on MACC are 50% and 10% larger, respectively.

Additionally, and although the pure dust lidar ratio used for CALIOP is 40 sr, sometimes the algorithm identifies part of the SAL as polluted dust, with an associated lidar ratio of 55 sr. This leads to an average lidar ratio larger than 40 sr, which in turn leads to a higher extinction and to a smaller systematic bias.

[Figure]

Extinction coefficient mean for June-July 2013

**Page 5, line 20, dust vertical distribution of what??? The vertical distribution is not a parameter!**

"Dust vertical distribution" will be replaced by "dust extinction vertical profile" in the revised version of this manuscript.

**Page5, lines 22-28: Are their CALIOP overflights over Barbados!... so that one could check the quality of the CALIOP profiles directly with Raman lidar profiles.**

A validation of CALIOP is out of the scope of this paper and there are already many good references about that. These references will be cited to provide an estimation of the uncertainties corresponding to the presented CALIOP retrievals.

**Page 8, line 16: Extinction coefficient and AOD are reported.., so no separation of dust and non-dust particle extinction? Just the total particle extinction! What lidar ratios are then used.., for the dust/marine mixture from about 1000-1500 m (SAL base) down to 400 m (below 400 m one may assume pure marine)**

**Section 3.3. It is a bit confusing, when just total particle extinction coefficients are compared without an approach to separate dust from marine.**

Unfortunately the MACC model dataset corresponding to the SALTRACE campaign period do not provide separation of the extinction by aerosol species.

**Page 10, line 2: Do you believe that ext coef. obtained from CALIOP is just 50 Mm-1 plus minus 20Mm-1 in the ABL, and all this is just marine aerosol?**

Our statement regarding the underestimation of the marine BL extinction by MACC doesn't refer to a specific aerosol type. Although normally the MBL extinction is dominated by the sea salt, there is certainly some dust and in some cases dust can dominate. A quantification of the contribution of each component to the total extinction is out of scope of this paper. A more detailed study on this can be found on this special issue (Groß et al., 2016, http://www.atmos-chem-phys-discuss.net/acp-2016-246/).

**Page 10, line 10: Who is right? MACC may be wrong, such layers reaching to 20 km height are not 'realistic'. And CALIOP cannot resolve such tiny aerosol traces.**

This is correct and explained in Pag. 10, lines 12-14. An independent verification of this feature is presented in the case study corresponding to the transfer flight between Cabo Verde and Brazil. In the model results corresponding to that flight aerosols are

visible above the SAL but not by the lidar. Independently of the accuracy of the DWL calibration and lidar ratio, in case of a real feature, some backscatter should have been detected by the DWL close to the Falcon. A discussion of the DWL sensitivity threshold will be added to the manuscript in order to further support our claims.

**Page 11, line18-20: I have many questions regarding the comparisons, without having a good answer, how to handle the quality of the comparisons. I would like to see a very careful discussion. One should again mention all the potential errors sources, and clearly state that the comparisons are combined with high uncertainties. As you know, CALIOP delivers only particle backscatter profiles. The Klett forward integration method is used. The solution can thus be very erroneous, especially at the end of the profile (lowest part of the atmosphere, i.e., in your cases, in the lowest part of the SAL and the layer below the SAL down to the ground). The uncertainty can easily be 30-50%, even larger. On the other hand, your DWL does not allow to retrieve rather accurate backscatter coefficients, too. The conversion to 532nm backscatter is combined with high uncertainties, and the further conversion to 532 nm extinction, as well. This introduces a systematic bias to the entire profile within a given layer (SAL, transition layer, marine MBL), and this bias is different for the different layers. How can you then state, for example, that MACC underestimates the extinction of the marine boundary layer and overestimates the extinction of the SAL? So I would like to see a very sensitive discussion in view of all the uncertainties on both sides, observations and modelling…**

We agree with the need of including a more detailed explanation of the magnitude of the uncertainties in the presented comparisons. On the other hand, we knew about the limitations of both, the DWL and the CALIOP retrievals. For that reason the statements regarding the aerosol extinction coefficient are **qualitative** (e.g. overestimation and underestimation).

As we already explained before, we claim that the model underestimates the extinction on the boundary layer and this is supported by the observations. We know more about the uncertainties of CALIOP than just a number. We know from the references you mentioned that CALIOP **underestimates**, especially in the boundary layer, the extinction coefficient due to the "accumulated effect" of the underestimated

lidar ratio for the Saharan dust. **The model is then underestimating a value which is already underestimated.**

As mentioned before, even after "correcting" the SAL extinction by multiplying the CALIOP results by 1.375 (55sr/40sr), MACC shows larger SAL AOD values than CALIOP. If instead of correcting CALIOP extinction by multiplying by 55/40 the mean underestimation indicated in Tesche et al. (2013) is used (1/0.858 - Table 5, summer), the difference is even larger.

A similar discussion applies to the observations discussed in Pag. 11, 18-20 performed by the DWL. In this case, we expect the DWL perform even better than CALIOP given the fact that we use a lidar ratio of 55 sr for Saharan dust. According to this case study, MACC overestimated the extinction in the SAL and underestimated the extinction in the BL.

**Page 12, lines 23-25, so if CALIOP cannot measure the 'artifact' (as produced by the MACC model) why do you not at least check the SALTRACE ground-lidar data, whether there was an aerosol layer in the upper free troposphere or not. Such layers at such great heights are clearly a large scale phenomenon... and should have been seen by the Barbados lidars, if existing.**

If existing, this aerosol layer should also be seen by the DWL. A discussion about the sensitivity threshold of the DWL will be included to support this affirmation.

Additionally, it has to be noted that the ability of ground-based lidars to detect thin aerosol layers with a extinction coefficient below 0.01 $km^{-1}$ on the upper troposphere will be also affected not only by the large distance but also by the attenuation introduced by the SAL and the boundary layer, which can exhibit optical depths as high as 0.55 during dust events.

**Page 13, section 4.3. This could be the central subsection of the entire paper. Here one could start with the comparison of DWL backscatter (and extinction profiles) with ones from the ground lidars. Afterwards, one could step forward with CALIOP and model output discussions.**

A comparison between the DWL and the ground-based lidar POLIS during the same day of the case study presented in section 4.3 is presented in Chouza et al. (2015) as part of the calibration method validation.

**The DWL extinction profiles… what lidar ratios did you use? For the dust layer, for the mixed dust/marin layer, for the marine layer...? Is that in agreement with ground lidar data? The ground-based lidars measure lidar ratios during darkness, and these lidar ratios are certainly valid hours before or later…, and thus applicable to the DWL observations.**

The lidar ratios used for the retrieval of extinction from the DWL were obtained from the POLIS ground-based lidar measurements. This is explained in Chouza et al. (2015). The lidar ratios used for the DWL inversion for the case studies presented in this paper are: 55 sr for dust, 35 sr for dust-marine mixtures, and 30 sr for marine.

**At the end I must say: It is quite strange to see an aerosol-related comparison paper on the basis of airborne Doppler lidar measurements, a lidar which does not measure backscatter coefficients..., furthermore based on CALIOP which does not measure extinction profiles, and the only lidars, delivering extinction profiles are not included in the paper**

This paper doesn't intend to be an "aerosol characterization paper". The objective of the paper is perform a general evaluation of the MACC model with emphasis on the long-range transport process and associated features like the AEJ and the AEWs. As part of this comparison, we use CALIOP and DWL extinction retrievals to evaluate the plume geometry and investigate **large** discrepancies between measurements and the model.

**I found a SAMUM 2 Raman lidar vs CALIOP intercomparison paper, Teshe et al. (JGR, 2013). You may not know that paper, but it should be referenced... More general, one should check and know all the SAMUM papers from 2009 and 2011 and provide proper referencing to all the SAMUM efforts done.**

Thank you for the reference. It is an interesting paper and will be certainly cited to support our statements.

**Do we need Figure 2 in this paper on CALIOP and MACC? Is that not already presented in wind-related SALTRACE papers?**

**Figure 3: Same question: : :**

This comparison was not yet published as part of wind-related SALTRACE papers. We think that an estimation of the DWL measurement accuracies of the horizontal wind are needed in order to support the evaluation of the dynamics of MACC presented in this paper.

**Figure 5: Why do you show this figure? You show two-month mean values, right? From all the CALIOP observations in June and July 2013? For proper comparison, the respective MACC results were averaged for the same CALIOP observational times within the two months?**

The observations are correct. CALIOP measurements were first downgraded to MACC resolution (aprox. 80 km). Then MACC was interpolated to the CALIOP overpass tracks and the results averaged. The purpose of the plot is to show the measured and modeled plume geometry as well as the large differences observed in BL extinction and some other differences compatible with the limited accuracy of CALIOP.

**Figures 5 a and b: Who is right? CALIOP or MACC? Who knows, I do not know? Because MACC is based on MODIS AOD, I would trust MACC. Because CALIOP needs lidar ratios they do not measure, and thus do not know... these AOD values are less trustworthy. What lidar ratio did they use for dust 40sr or 55sr? CALIOP suffers from multiple scattering effects in dust. Is that taken into account. MS leads to underestimation of AOD.**

This paper doesn't pretend to provide an explanation to the CALIOP/MODIS differences. A reference discussing such differences is cited in the manuscript (Kim et al. (2013)). The AOD comparison is presented to analyze meridional displacements of the SAL "center" (which is not very sensitive to a bias in the extinction).

**Figures 5 g and h.... Again, because the forward integration Klett method has to be used (very sensitive to uncertainties in the lidar ratio input profile), I do not trust the CALIOP values in the MBL and the mixing zone above MBL. Again,**

**who is right? There is no answer! Should be critically discussed! The only answer could be given by the SALTRACE ground lidars.**

This point is already discussed.

**Figure 8: Again! Who is right (in figs 8 a and b). DWL cannot measure extinction. This is a wrong statement. The lidar can even not measure backscatter! It needs help by 'real' aerosol lidars. So I am always puzzled, what the basic and essential goal of this paper is…? Yes, the wind data comparison is very attracting, very convincing! This is the strongest part of the paper. MACC obviously does a good job.**

**Figure 9: Again.., the only reliable information (in a and c) is the observation of dust layering.**

We agree with the reviewer in the incorrect use of the term measurement, this will be corrected in the revised version of the manuscript. Nevertheless, DWLs can retrieve extinction under certain assumptions and conditions. This leads to relative large uncertainties (compared to a Raman lidar) which have to be taken in account. Nevertheless, the impact of the uncertainty depends on the intention of the derived conclusions. We extract conclusions from these retrievals taking in account these limitations. For that reason our statements regarding the aerosol measurements are also limited and in many cases qualitative. On the other hand, the higher quality of our wind measurements allows us to extract more accurate conclusions.

**Figure 10: Is this figure needed. Ok, convincing! MACC does a good job! If that is an important finding, leave it in. If not, remove the figure.**

It is certainly an important finding taking into account that one of the objectives of the publication is to evaluate different aspects of MACC.

**Figure 11: At least for 11 July, I expected to see a broader view on the aerosol situation (conditions). Here, I would like to see the other 'real' SALTRACE aerosol lidar observations… in comparison with the DWL observations.**

Such comparison is already included in Chouza et al. 2015.

**I am always confused by the fact, that this will be a contribution to a SALTRACE Special Issue, but the special issue aspect, integration of all available measurements to design a complete aerosol picture, is only poorly given. It seems to me that authors need publications and do not really take care and the time to look at all available data.**

To give a "complete aerosol picture", ranging from accurate optical and microphysical properties to large scale transport patterns, was not intended with this single paper. A "complete picture" is probably out of scope of any single paper. The objective of this paper is to contribute to a fraction of the whole picture, with emphasis on the transport process and the evaluation of the current modeling capabilities to reproduce it. We consider the "complete picture" of the complex dataset and modelling efforts resulting from SALTRACE will emerge from the number of papers published and to be published in this special issue.